# Federated Continual Learning Goes Online: Uncertainty-Aware Memory Management for Vision Tasks and Beyond

**Giuseppe Serra**[1,2] **& Florian Buettner**[1,2,3]

`serra@med.uni-frankfurt.de` & `florian.buettner@dkfz-heidelberg.de`

## Abstract

Given the ability to model more realistic and dynamic problems, Federated Continual Learning (FCL) has been increasingly investigated recently. A well-known problem encountered in this setting is the so-called *catastrophic forgetting*, for which the learning model is inclined to focus on more recent tasks while forgetting the previously learned knowledge. The majority of the current approaches in FCL propose generative-based solutions to solve said problem. However, this setting requires multiple training epochs over the data, implying an offline setting where datasets are stored locally and remain unchanged over time. Furthermore, the proposed solutions are tailored for vision tasks solely. To overcome these limitations, we propose a new approach to deal with different modalities in the *online* scenario where new data arrive in streams of mini-batches that can only be processed once. To solve catastrophic forgetting, we propose an uncertainty-aware memory-based approach. Specifically, we suggest using an estimator based on the Bregman Information (BI) to compute the model's variance at the sample level. Through measures of predictive uncertainty, we retrieve samples with specific characteristics, and – by retraining the model on such samples – we demonstrate the potential of this approach to reduce the forgetting effect in realistic settings while maintaining data confidentiality and competitive communication efficiency compared to state-of-the-art approaches.

## 1 Introduction

In recent years, federated learning (FL) (McMahan et al., 2017) has received increasing attention for its ability to allow local clients work towards the same objective without the need for sharing limited and sensitive information. However, despite the benefits provided by the privacy-preserving collaborative training, the assumptions of the standard scenario are far from realistic (Babakniya et al., 2024). For example, the standard static-single-task framework has very limited practical applicability in real-world cases where local clients often need to continuously learn new tasks. Let us consider the problem of classifying new COVID-19 variants as illustrative use-case. Given the evolving nature of the virus, new variants (i.e., new classes) arise over time. In this context, healthcare facilities which collaborate using FL would fail since the standard setting does not consider the dynamic increment of new classes. For this reason, a new paradigm was recently introduced to model more complex dynamics: federated continual learning (FCL) (Yoon et al., 2021a; Dong et al., 2022). This new paradigm takes the global-local communication and privacy-preserving abilities enabled by FL and combines them with the Continual Learning (CL) ability of learning different tasks sequentially over time. However, FCL not only shares the characteristics at the intersection of FL and CL, but also inherits their respective challenges. In particular one of the most prominent problems in CL, the so-called *catastrophic forgetting* (CF) for which the model is prone to suffer from significant performance degradation on older tasks. In CL, this problem is addressed in different ways ranging from memory-based approaches (Yoon et al., 2021b; Kumari et al., 2022; Hurtado et al., 2023) to generative methods (Shin et al., 2017; Lesort et al., 2019) or to regularization techniques (Lee et al., 2017; Aljundi et al., 2018; Chaudhry et al., 2018b; He & Jaeger, 2018). In the context of FCL, instead, the majority of the proposed approaches exploit generative models at

---

[1]Goethe University Frankfurt,[2]German Cancer Consortium (DKTK),[3]German Cancer Research Center (DKFZ)

both local and global level to trace and encode the past information and generate synthetic instances which faithfully mimic previous history whilst preserving data privacy. A substantial drawback of such generative approaches is that they are tailored to image data and it is not clear how they could translate to other data modalities. Importantly, such generator-based approaches also imply an *offline* setting where task data are collected before training and remain unchanged over time: local generators require large training datasets of complete tasks and similarly, more recently proposed generative models based on data distillation can only be trained at the end of each task (Babakniya et al., 2024).

In realistic conditions however, all data from the current task may not be available for collection at the same time but may rather arrive in small chunks sequentially. Furthermore, given the ubiquity of smart and edge devices with limited capabilities (e.g., wearable devices), there is a need for updating the learning model with the new incoming data to minimize memory overload and communication bandwidth (Ma et al., 2022). This problem of learning from an *online* stream of data is well investigated in CL while it remains largely unexplored in FCL.

Inspired by the problem of training models over *online* streams of data, we first formalise a new scenario to tackle the online problem in FCL (online-FCL). In this new scenario, we assume each client to learn from a stream of data where new data arrive in mini-batches which can only be processed once. As such, in line with the definition of online-CL, the model is updated with high frequency (Soutif-Cormerais et al., 2023). Then, taking inspiration from the most popular solutions in online-CL, we introduce memory buffers to alleviate CF at local level. More in detail, we propose an uncertainty-based memory management where data points are stored in local buffers according to their predictive uncertainty. Intuitively, predictive uncertainty provides a glimpse of the samples' location in the decision space; samples with low uncertainty are the most representative ones for the respective class, while samples with high uncertainty represent data points that are close to the decision boundary and/or outliers. Thus, via estimates of predictive uncertainty, we can store in the memory samples with desired properties. Here, we propose to quantify uncertainty by directly estimating a generalized variance term from the loss function, which can be interpreted as a measure of epistemic uncertainty. To this end, we leverage a recently proposed bias-variance decomposition of the cross-entropy loss (Gruber & Buettner, 2023) and estimate the Bregman Information (BI) as the variance term in logit space.

The uncertainty-based memory management makes our approach flexible in terms of data modality. In fact, independent of the data modality (e.g., images, texts), our solution allows us to compute uncertainty estimates and thus populate the memory accordingly. This is in contrast to most of the current solutions in FCL which are limited to vision tasks and the offline setting.

In the last part of the paper, we demonstrate the ability of the proposed approach to reduce the forgetting effect in different scenarios. In the first part of the experiments, we evaluate our approach on CIFAR-10 and CIFAR-100 (Krizhevsky et al., 2009), two standard datasets used in this context. The goal is to understand how the proposed (epistemic) uncertainty estimate based on the Bregman Information performs in comparison with other standard estimates of overall model uncertainty under a memory-based regime. Then, departing from the common evaluation pipeline that would involve datasets like EMNIST (as in Qi et al. (2023); Wuerkaixi et al. (2024)) or larger datasets from the same naturalistic domain (e.g., ImageNet – as in Qi et al. (2023); Babakniya et al. (2024)), we validate our results on more probing real-world datasets from the medical domain. Finally, to showcase the ability of our approach to work with different data modalities, we test our findings on text classification tasks.

The contributions of this work can be summarized as follows:

- We propose and formalise a novel framework to tackle the FCL problem in the online setting which is largely overlooked in the literature.
- We highlight the limitations of current state-of-the-art generative-based solutions to work in the online setting and show empirically their inefficiency in the experimental section.
- We propose a memory-based solution that employs an alternative estimate for predictive uncertainty – which stems directly from a bias-variance decomposition of the cross-entropy loss for classification tasks – to populate the memory.
- We demonstrate the efficacy of our method in more realistic scenarios including datasets from different domains, with imbalance, and with different modalities.

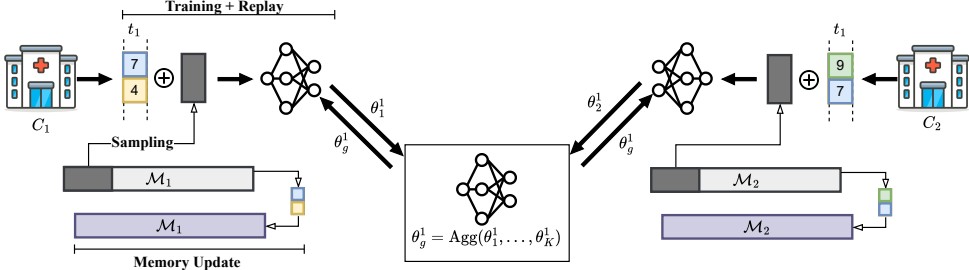

Figure 1: Schematic overview of the proposed online-FCL scenario. Each client $\mathcal{C}_k$ receives their data as a stream of batches. Each batch $b_k^t$ is used for training and for updating the local memory $\mathcal{M}_k$. Once the current batch is processed, the client receives a new batch and continues their training without the possibility of retraining on previously seen batches.

## 2 RELATED WORK

**Continual Learning.** There are three types of incremental learning (IL) (van de Ven et al., 2022); task-, domain-, and class-IL. In task-IL, training and testing data include explicitly task IDs. In this unrealistic scenario, the learning algorithm knows which task needs to be completed. In the domain-IL setup, instead, the classification problem remains the same while the input distribution (or domain) shifts over time. For instance, an example of domain drift appears when a model for detecting even/odd digits in images learns first $1s$ and $2s$ and then $4s$ and $5s$. Lastly, class-IL scenarios represent the most realistic setting where the model continuously learns from an increasing number of classes. CL problems can also be classified based on whether the data can be stored and viewed multiple times (*offline*) or whether they can be processed once (*online*) (Mai et al., 2022).

**Class-Incremental Learning.** Methods for class-IL can be divided into several categories (Mai et al., 2022) as follows; a) *Regularization techniques* alter the model parameter updates by adding penalty terms to the loss function (Lee et al., 2017; Aljundi et al., 2018), adjusting parameter gradients during optimization (Chaudhry et al., 2018b; He & Jaeger, 2018), or employing knowledge distillation (Rannen et al., 2017; Wu et al., 2019); b) *Memory-based techniques* exploit a fixed-size buffer containing samples from past tasks for replay (Aljundi et al., 2019; Chaudhry et al., 2019) or regularization (Nguyen et al., 2018; Tao et al., 2020); c) *Generative-based techniques* involve training generative models that can produce pseudo-samples mimicking data from the past (Shin et al., 2017; Lesort et al., 2019); d) *Parameter-isolation-based techniques* assign different model parameters to each task. This can be done either by activating only the relevant parameters for each task (Fixed Architecture) (Mallya & Lazebnik, 2018; Serra et al., 2018) or by adding new parameters and keeping the existing ones unchanged (Dynamic Architecture) (Aljundi et al., 2017; Yoon et al., 2018). In online-CL, where data arrive in single mini-batches and the model is updated frequently, rehearsal-based methods are favoured over more complex solutions, like generative methods, because they offer greater flexibility and require less training data and computational time (Mai et al., 2022).

In the context of memory-based techniques, the main question is how to optimally manage the memory. In order to reduce catastrophic forgetting, samples in the memory should be representative of their own class, discriminative towards the other classes, and informative enough for the model to recall the information about the old classes. In the literature, many conflicting strategies can be found. Some of them suggest to use the most representative samples (Yoon et al., 2021b; Hurtado et al., 2023), while others consider the samples near the decision boundary as the most useful to reduce CF (Kumari et al., 2022).

**Federated Continual Learning.** The intersection of federated learning and continual learning has only been investigated very recently. One of the first papers in this direction is Yoon et al. (2021a). The approach focuses on the less challenging task-IL scenario, where the task ID information is required during inference and testing. In Dong et al. (2022), the authors introduce the federated class-IL (FCIL) problem which tends to model more realistic scenarios. The clients have access

to a large memory buffer and can share perturbed prototype samples with the global server, which differs with the standard FL setting where only the parameters of the local models are shared with the server (Babakniya et al., 2024). Ma et al. (2022) uses knowledge distillation on both local and global levels via unlabeled surrogate datasets. Other works (Hendryx et al., 2021; Qi et al., 2023; Babakniya et al., 2024; Wuerkaixi et al., 2024) pose their attention on the FCIL problem without the use of memory replay data. Hendryx et al. (2021) focus on few-shot learning and allows overlapping classes between tasks. Qi et al. (2023) instead, introduces the constraint of non-overlapping classes for intra-client tasks. The work is based on generative replay where clients train a discriminator and a generator locally. At the same time, at each communication round, the server performs a consolidation step and generates synthetic data using the locally trained generators. Following a similar principle, Babakniya et al. (2024) presents a generative model which is trained by the server in a data-free manner. Another FCIL scenario is considered in Shenaj et al. (2023), where tasks can arrive asynchronously at each client. The problem is tackled via a combination of prototype-based learning, representation loss, and stabilization of server aggregation.

Here we briefly summarise the identified limitations of such generator-based approaches. As anticipated before, they assume to work in the offline setting where static datasets are collected in advance and stored locally. This allows them to be trained over the same task dataset many times which is needed to reach convergence and to learn meaningful patterns in order to generate informative synthetic images. Finally, the storage of the whole task datasets and the generator models may be unfeasible in resource-limited devices.

## 3 OUR APPROACH: O-FCL

### 3.1 ONLINE-FCL: PROBLEM FORMULATION

Following the notation proposed in Qi et al. (2023), we assume to have a set $\mathcal{C} = \{\mathcal{C}_1, \ldots, \mathcal{C}_n\}$ of $n$ different clients. Each client $\mathcal{C}_k$ keeps its private data $\mathcal{D}_k = \{D_k^1, \ldots, D_k^t\}$ with its corresponding sequence of tasks $\mathcal{T}_k = \{t_k^1, \ldots, t_k^{\hat{t}}\}$. For each client $k$ and task $t$, we have an associated dataset $D_k^t = \{(x_i, y_i)\}_{i=1}^{n_k^t}$ with $x_i$ an input sample, $y_i$ the corresponding class label, and $n_k^t$ the number of training samples. In the proposed *online* setting, we assume that samples for each task $t$ come gradually in a stream of mini-batches $b_k^t = \{(x_i, y_i) \in D_k^t\}_{i=1}^{bs}$ which can only be seen once. Similar to Qi et al. (2023), we assume non-overlapping classes for intra-client tasks. In other words, the clients can see a specific class only once during the training. At each communication round $r$, the client trains the local model on its own data and shares the local parameters $\boldsymbol{\theta}_k^r$ to update the global model parameters $\boldsymbol{\theta}_g^r$. Since data points can be processed only once, in order to exploit the information from the incoming data points at its best, communications with the server are performed when a single mini-batch or multiple consecutive mini-batches are processed. This is considerably different from the standard FCL scenario, where multiple iterations and communication rounds are performed for the *whole task dataset*. Following a popular trend in FCL, we also focus on the FCIL scenario which represents the most realistic and challenging case.

### 3.2 METHODOLOGY

As briefly anticipated above, the online nature of the newly introduced problem poses new challenges compared to the standard case. For this, we propose a memory-based approach on the client-side to alleviate catastrophic forgetting. The motivation is threefold: 1) the most effective solutions in online-CL are approaches with memory buffers; 2) given the online nature of the problem, generative-based approaches would be limited as they require many iterations over the same dataset; 3) compared with generative-based solutions, we only store a small amount of data reducing the overall overhead on the local device.

In the following subsections, we describe in more detail the characteristics of our approach at the client and server levels. For the client side, we outline our uncertainty-aware memory management detailing the properties of the employed uncertainty estimate and its advantages compared to standard scores. For the server side, we detail the adjustments done to improve the communication and parameter averaging effectiveness in the online scenario. A summary of the complete training procedure is reported in Algorithm 1 (see Appendix A.11).

Figure 2: Illustration of the different aspects of uncertainty captured by model confidence scores and Bregman Information (BI). Close to the decision boundary we can notice how, due to the high aleatoric uncertainty, data points have low-confidence scores. Contrarily, due to high density of observed data, there is a low uncertainty about the data generating process resulting in a low BI.

### 3.2.1 CLIENT-LEVEL

**Memory management.** For each client $k$, we introduce a fixed-size memory buffer $\mathcal{M}_k$. Similar to Chrysakis & Moens (2020), the memory population strategy is based on a class-balanced update, which is crucial to consider in case the stream of data is highly imbalanced. In fact, if classes are not equally represented in the memory, sampling from the memory may further deteriorate the predictive performance of the framework for under-represented classes. Differently from the populating strategy proposed in Chrysakis & Moens (2020), the criteria to decide which samples to keep in the memory is not random, but based on predictive uncertainty estimates in order to intentionally store samples with desired characteristics for each class. There are different ways to select the samples. We can decide to store a) the class-representative data points by selecting the least uncertain ones for each class (*bottom-k*); b) the easiest-to-forget samples by sampling the ones with high uncertainty (*top-k*).

From the second task on, we need a strategy for sampling from the memory a subset of data points used for replay (*replay set*). Since the memory represents a prototypical set of data points for each class, we assume that a random sampling is sufficient to extract informative data points from the memory. Following the standard practice, the number of samples in the replay set is equal to the batch size. It is important to note that the memory may already contain samples from the current task. In such cases, samples from the current task are excluded from the sampling process to ensure that the focus remains solely on those belonging to past tasks.

**Predictive uncertainty estimation in the logit space.** Different measure of predictive uncertainty can capture distinct aspects of a model's irreducible aleatoric uncertainty (inherent in the data) and its epistemic uncertainty (that stems from the finite training data and can be reduced by gathering more data). In online CL, the most commonly used measures are derived directly from the confidence scores of the model and mostly capture the irreducible aleatoric uncertainty (Wimmer et al., 2023). Here, we hypothesize it may be more beneficial for the model to replay instances which are representative in the sense that there is low uncertainty about the data generating process (we refer to this as low epistemic uncertainty, while acknowledging that there exist varying definitions). To not rely on specialized Bayesian models, we leverage a recently proposed bias-variance decomposition of the cross-entropy loss and compute a generalized variance term directly from the loss (Gruber & Buettner, 2023). Such bias-variance decomposition decomposes the expected prediction error (loss) into the model's bias, variance, and an irreducible error (noise term). The latter is related to aleatoric uncertainty, whereas the variance term can directly be related to epistemic uncertainty (Gruber et al., 2023). This is the first attempt of using this decomposition for memory management in FCL contexts.

Gruber & Buettner (2023) have recently shown that a bias-variance decomposition of cross-entropy loss gives rise to the Bregman Information as the variance term and measures the variability of the prediction in the logit space. We illustrate the different aspects of uncertainty captured by confidence scores and BI respectively in Figure 2. For example, data points close to the decision boundary have a low confidence score due to the inherently high aleatoric uncertainty; in contrast, due to the high density of observed data, there is actually a low uncertainty about the data generating process (DGP), resulting in a low BI (low epistemic uncertainty). Outliers far away from the decision boundary can have a high confidence score, but will have a high BI due to the high uncertainty regarding the DGP.

We hypothesize that it is samples with a low BI that are most useful for replay in online-FCL. To populate the memory with representative samples, we therefore propose using an uncertainty estimator based on Bregman Information (BI) (Gruber & Buettner, 2023). The authors demonstrate that BI at the sample level can be estimated through deep ensembles or test-time augmentation (TTA). However, to reduce the computational overhead at the local level, we employ TTA for computing the estimations. Let us consider the problem of multi-class classification (as in our case), where the standard loss is represented by the cross-entropy. Considering a set $P$ of perturbations for a given data point $x$, we can compute the variance term of the classification loss $u(x)_{BI}$ as follows:

$$u(x)_{BI} = \frac{1}{P} \sum_{i=1}^{P} \text{LSE}(\hat{z}_i) - \text{LSE}\left(\frac{1}{P} \sum_{i=1}^{p} \hat{z}_i\right), \tag{1}$$

where $\hat{z}_i \in \mathbb{R}^c$ represent the logit predictions and $\text{LSE}(x_1, \ldots, x_n) = \ln \sum_{i=1}^{n} e^{x_i}$ the *LogSumExp* (LSE) function respectively. Intuitively, a large value of $u(x)_{BI}$ means that the logits predicted across the perturbations vary significantly, suggesting a high uncertainty of the prediction and the DGP at this point of the input space. The use of this estimator is motivated by the fact that, in comparison with other uncertainty scores such as entropy, smallest margin, or least confidence, there is no information loss in the estimation step. In fact, if we inspect these alternative metrics (see Appendix A.1 for the equations), we can notice that they either need a normalization step to move the logits in the probability space or rest on the largest activation value only. Furthermore, as reported in Gruber & Buettner (2023); Ovadia et al. (2019), and Tomani & Buettner (2021), common confidence scores are reliable only in case of well-calibrated models. In contrast, the BI-based estimation of the epistemic uncertainty is meaningful also under distribution shift and able to identify robust and representative samples (Gruber & Buettner, 2023).

### 3.2.2 GLOBAL-LEVEL

**Communication rounds.** Following the standard assumptions of the federated scenario, we allow local clients to share the model parameters solely. However, the proposed online framework poses additional challenges compared to the standard case. In the standard scenario, the communication round is performed at the end of several iterations over the same task dataset. This implies that the model has probably reached convergence and can effectively share the learned information during the communication round. In our case, a few new samples are available at each step and, to keep the model up-to-date, the communication round cannot be performed at the end of the task only. For this reason, as anticipated in Section 3.1, communications with the server are performed when a single mini-batch or multiple consecutive mini-batches are processed. Given the instability of the model when a new task starts, to ensure an effective and efficient information sharing, we propose to set a *burn-in* period. During this period, the local model learns independently for a certain number of batches without sharing and receiving information from the others. This guarantees that, when the local client starts to be involved in the communication rounds, the model has effectively learned relevant information on the current task. Additionally, since every parameter update degrades the predictive performance at local level (Qi et al., 2023), we propose to limit the number of communications rounds per task. Instead of performing a model averaging every time a batch is processed, we let the local models learn for $q$ consecutive mini-batches before actively participating to the communication round.

**Parameter averaging.** Although clients work on the same task, they may receive data coming from different classes where some of them are over-represented compared to others. In this case, the local parameters collected for parameter averaging can be *biased* towards the most common classes. For this, we suggest to first create an aggregated model for each class available in the current round, and then compute $\boldsymbol{\theta}_g^r$ using the class-based models just created. This way, the current classes contribute equally during the updates of the global parameters. In case all the clients receive data containing classes equally distributed, the parameter averaging results in the standard computation, like e.g., FedAvg (McMahan et al., 2017). Importantly, if sharing the class information is not possible, we can rely on standard averaging strategies (e.g., FedAvg or FedProx (Li et al., 2020)) without hampering the performance since our methodology is flexible on this matter (see Table 9). Finally, once the new global parameters $\boldsymbol{\theta}_g^r$ are computed, to avoid a drastic change between the old parameters and the new ones, following Shenaj et al. (2023), we propose to average the newly computed parameters at round $r$ with the previous parameters computed at round $r - 1$.

## 4 EXPERIMENTS

### 4.1 DATASETS AND SETTINGS

**Datasets.** To understand the behaviour of different algorithms in the online-FCL scenario, we use two datasets commonly used in the literature, namely, CIFAR10 and CIFAR100 (Krizhevsky et al., 2009), and TinyImageNet (Le & Yang, 2015). We randomly assign a set of classes to 5 tasks (10 for TinyImageNet) and we split the task data evenly among the clients such that the task sequence $T_k^t$ for task $t$ and client $k$ does not share any data with the other clients. In this way, since the class assignment per task is different every time, we can identify the strategy providing greater flexibility and effectiveness irrespective of the composition of the tasks. Then, as anticipated in the introductory part, we assess the performance under more difficult and realistic conditions. Instead of using datasets that, to some extent, share similar characteristics with CIFAR (i.e., ImageNet) or represent unrealistic tasks (such as EMNIST), we validate our results on datasets for biomedical image analysis. Apart from the change in the domain (which in turn means different backgrounds in the images, different statistics, etc.), these datasets pose an additional challenge compared to the standard benchmark datasets, i.e., data imbalance. To reflect realistic conditions where recent tasks contain fewer data points than the older ones since the time to collect them is shorter, we assign classes to tasks based on the class size. We decide to focus on two biomedical datasets annotated by expert clinical pathologists; the colorectal cancer histology (CRC-Tissue) dataset (Kather et al., 2019; Yang et al., 2023) containing images divided in 8 classes of hematoxylin–eosin (HE)–stained slides taken from patients with colorectal cancer (CRC), and the kidney cortex cells (KC-Cell) dataset containing 8 classes of human kidney cortex tissue sections (Ljosa et al., 2012; Yang et al., 2023). Finally, to evaluate our approach on text classification tasks, we employ three different datasets: the 20NewsGroups dataset (Lang, 1995), DBPedia (Zhang et al., 2015), and Yahoo Answers (Zhang et al., 2015). As for vision tasks, we randomly assign classes to tasks every run. Statistics and more details about the used datasets can be found in the supplementary material (Appendix A.3).

**Experimental settings.** Given the novelty of the online setting, there are no direct competitors which we could refer to. We therefore investigate the most successful algorithms from the online-CL and the FCL literature. We compare our approach with standard baselines for FL, i.e., FedAvg (McMahan et al., 2017) and FedProx (Li et al., 2020), a standard memory-based approach for online-CL, namely Experience Replay (ER) (Chaudhry et al., 2019), and two state-of-the-art approaches for FCL with generative replay, namely MFCL (Babakniya et al., 2024) and FedCIL (Qi et al., 2023). Our decision to use ER is based on the fact that, despite its simplicity, it is surprisingly competitive when compared with more sophisticated and newer approaches as shown in recently conducted empirical surveys (Soutif-Cormerais et al., 2023). For FCL, we are particularly interested in MFCL because its data-free solution trains the generator on the server side. As such, we believe it may be potentially feasible in the online scenario, in contrast to other approaches such as FedCIL that train local generators at the client-side (Qi et al., 2023). In addition to the standard ER, we also include the class-balanced version (CBR) presented in Chrysakis & Moens (2020). Finally, to show the competitiveness of $u(x)_{BI}$, we also consider other uncertainty scores. In particular, least confidence (LC), margin sampling (MS), ratio of confidence (RC), and entropy (EN) (Shannon, 1948; Campbell et al., 2000; Culotta & McCallum, 2005).

In all the experiments for image classification, we employ a slim version of Resnet18 (He et al., 2016) – as done in previous works in online-CL (Kumari et al., 2022; Hurtado et al., 2023; Soutif-Cormerais et al., 2023) –, and use the SGD optimizer with a learning rate of $0.1$. For text classification, we employ a simple MLP with a single fully connected hidden layer (512 units) and Adam optimizer with a learning rate of $0.01$. To generate the perturbed inputs for estimating predictive uncertainty, we use two different strategies according to the different data modalities. For vision tasks, the perturbations are generated via standard augmentation. The list of augmentations used in our experiments is provided in Appendix A.2. For the natural language experiments, we leverage recent progress in foundation models and first create general-purpose latent representations of the input texts via a pre-trained sentence embedder. In particular, we employ *e5-small-v2* (Wang et al., 2022) (384 dimensions) via HuggingFace (Wolf et al., 2020). Then, for perturbing the generated vector representations, we add Gaussian noise from $\mathcal{N}(0, 0.1)$ to each latent dimension.

Following standard practice, we set the batch size equal to $10$. The memory size was set to different values for each dataset in order to examine the performance of various memory configurations,

Table 1: Comparison of average last accuracy (A) and last forgetting (F) on CIFAR10 (5 tasks).

| Score | | M=200 | | M=500 | | M=1000 | |
|---|---|---|---|---|---|---|---|
| ER | | 20.74 ±2.53 | | 27.90 ±2.03 | | 33.64 ±0.72 | |
| | | 48.48 ±3.95 | | 30.18 ±2.69 | | 24.3 ±1.73 | |
| CBR | | 22.82 ±1.55 | | 24.17 ±3.23 | | 32.67 ±1.84 | |
| | | 46.79 ±2.92 | | 28.54 ±2.22 | | 23.62 ±2.63 | |
| | | Top | Bottom | Top | Bottom | Top | Bottom |
| LC | A(↑) | 21.58 ±1.56 | 20.92 ±0.70 | 29.49 ±1.46 | 27.38 ±1.68 | 36.14 ±2.44 | 31.94 ±2.44 |
| | F(↓) | 50.78 ±1.96 | 49.78 ±3.16 | 36.13 ±2.54 | 30.17 ±2.52 | 21.97 ±4.06 | 29.53 ±4.46 |
| MS | A(↑) | 22.34 ±1.04 | 21.88 ±1.28 | 29.72 ±1.98 | 28.96 ±1.54 | 34.42 ±0.91 | 32.28 ±1.20 |
| | F(↓) | 49.38 ±3.09 | 47.20 ±4.97 | 35.16 ±3.20 | 28.07 ±2.71 | 29.58 ±2.73 | 28.64 ±4.34 |
| RC | A(↑) | 21.80 ±1.65 | 22.75 ±1.84 | 28.36 ±2.07 | **31.17** ±1.75 | 35.08 ±1.11 | 28.98 ±2.26 |
| | F(↓) | 56.91 ±1.90 | 45.05 ±2.11 | 36.47 ±2.84 | 31.86 ±4.16 | 26.10 ±1.76 | 28.39 ±2.95 |
| EN | A(↑) | 21.15 ±1.37 | 20.14 ±1.41 | 28.63 ±0.86 | 26.48 ±1.59 | **36.25** ±1.22 | 26.42 ±1.49 |
| | F(↓) | 53.30 ±2.97 | 55.58 ±3.49 | 36.70 ±2.21 | 35.85 ±3.17 | 25.52 ±3.01 | 34.64 ±3.40 |
| BI | A(↑) | 21.31 ±1.46 | **24.89** ±0.83 | 26.57 ±0.83 | 27.84 ±2.31 | 35.12 ±2.51 | 35.83 ±2.60 |
| | F(↓) | 54.65 ±1.83 | **35.77** ±4.13 | 42.86 ±3.81 | **24.59** ±3.16 | 27.00 ±2.99 | **19.07** ±2.17 |

including both large and small buffers. The burn-in period is set to 30. A communication round with the server is performed after $q = 5$ mini-batches. The number of clients is set to 5. For parameter averaging, we employ FedAvg (McMahan et al., 2017). Given the flexibility of the approach, as mentioned before, the model averaging strategy can be changed as desired. For evaluation, following recent works (Yoon et al., 2021a; Wuerkaixi et al., 2024), we use the average last accuracy (A) and average forgetting (F) (see appendix A.5 for the definitions). All the experiments are run on three different random seeds. For each dataset, experiments were run on a Linux machine using a single Quadro RTX 5000 and 16 GB RAM.

## 4.2 EMPIRICAL RESULTS

From the experiments conducted on the CIFAR datasets, we can observe that the proposed approach is able to reduce CF consistently across different memory sizes and class-per-task assignments. The results in Tables 1 and 2 suggest that storing the class-representative samples (i.e., the least uncertain data points for each class) provides a benefit in terms of predictive performance gain and forgetting reduction. Our findings are substantiated in the classification tasks for biomedical images. Although the images originate from another domain, Table 3 confirms that in terms of CF, BI outperforms the considered memory-based baselines in most of the cases. Crucially, we also show the ability of our approach to work with imbalanced real-world data. Table 5 and Figure 3 summarise our findings; in comparison with both memory-based and generative-based methods, our simple-yet-elegant approach is able to perform consistently across tasks (left plot) and reduce CF on different datasets (right plot). Finally, we demonstrate that our method outperforms baselines also on non-vision data. Table 4 reports the results on the textual classification datasets. Here we find that the difference in terms of predictive accuracy is less marked. This is because the baselines have higher accuracy on the last tasks and poor performance on the first ones, while in our case the accuracy is kept more uniform across the tasks (reflected in a lower average forgetting) as shown in the left plot in Figure 3. Note that the use of generative baselines is restricted to image data only. Results for TinyImageNet are reported in Appendix A.4 - Table 7.

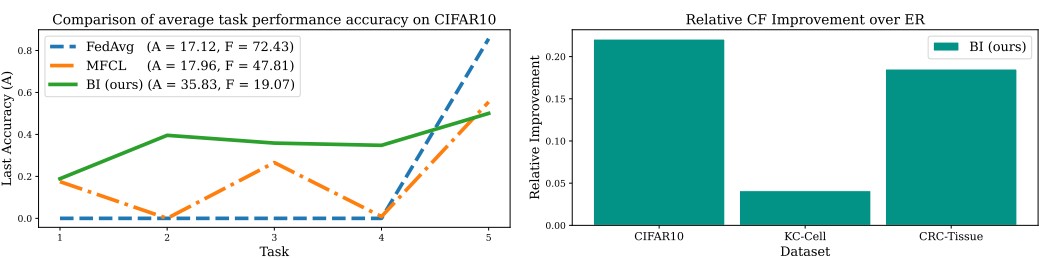

Figure 3: (left) Average accuracy per task in comparison with FL (FedAvg) and generative-based FCL baselines (MFCL); (right) Relative CF improvement over the CL baseline (ER).

Table 2: Comparison of last accuracy (A) and last forgetting (F) on CIFAR100 (5 tasks).

| Score | | M=1000 | | M=2000 | |
|---|---|---|---|---|---|
| ER | | $10.33 \pm 0.58$ | | $10.95 \pm 0.4$ | |
| | | $6.89 \pm 1.53$ | | $9.12 \pm 0.96$ | |
| CBR | | $13.38 \pm 0.58$ | | $11.41 \pm 0.36$ | |
| | | $31.66 \pm 0.30$ | | $9.37 \pm 1.56$ | |
| | | Top | Bottom | Top | Bottom |
| LC | A($\uparrow$) | $12.71 \pm 0.96$ | $13.43 \pm 1.00$ | $13.50 \pm 0.81$ | $14.09 \pm 0.47$ |
| | F($\downarrow$) | $8.03 \pm 1.88$ | $6.95 \pm 0.91$ | $7.99 \pm 1.34$ | $7.75 \pm 1.87$ |
| MS | A($\uparrow$) | $13.34 \pm 1.71$ | $13.77 \pm 1.31$ | $13.83 \pm 0.56$ | $14.26 \pm 0.72$ |
| | F($\downarrow$) | $7.48 \pm 2.04$ | $6.84 \pm 1.60$ | $8.04 \pm 0.99$ | $8.01 \pm 1.76$ |
| RC | A($\uparrow$) | $13.30 \pm 1.47$ | $13.76 \pm 1.24$ | $13.28 \pm 1.23$ | $13.83 \pm 0.69$ |
| | F($\downarrow$) | $8.20 \pm 1.91$ | $6.97 \pm 1.12$ | $7.52 \pm 1.70$ | $7.70 \pm 1.58$ |
| EN | A($\uparrow$) | $12.77 \pm 1.00$ | $12.77 \pm 0.71$ | $13.81 \pm 1.00$ | $13.93 \pm 0.76$ |
| | F($\downarrow$) | $8.11 \pm 1.14$ | $7.90 \pm 1.20$ | $8.10 \pm 1.51$ | $7.50 \pm 1.17$ |
| BI | A($\uparrow$) | $12.89 \pm 0.97$ | $\mathbf{14.04} \pm 0.62$ | $13.72 \pm 0.70$ | $\mathbf{14.31} \pm 0.76$ |
| | F($\downarrow$) | $7.54 \pm 1.43$ | $\mathbf{6.73} \pm 1.76$ | $8.06 \pm 1.49$ | $\mathbf{6.96} \pm 1.40$ |

Table 3: Comparison of average last accuracy (A) and last forgetting (F) on (left) CRC-Tissue (4 tasks), and (right) KC-Cell (4 tasks).

| CRC-Tissue | | | | | |
|---|---|---|---|---|---|
| Score | | M=80 | | M=120 | |
| ER | | $49.79 \pm 1.93$ | | $50.68 \pm 2.69$ | |
| | | $20.31 \pm 8.76$ | | $23.96 \pm 4.78$ | |
| CBR | | $46.70 \pm 1.34$ | | $49.38 \pm 1.01$ | |
| | | $33.69 \pm 3.16$ | | $19.19 \pm 6.86$ | |
| | | Top | Bottom | Top | Bottom |
| LC | A($\uparrow$) | $47.66 \pm 2.52$ | $56.02 \pm 4.89$ | $56.49 \pm 2.21$ | $58.12 \pm 2.94$ |
| | F($\downarrow$) | $41.42 \pm 3.64$ | $13.25 \pm 4.91$ | $26.65 \pm 6.30$ | $11.13 \pm 3.12$ |
| MS | A($\uparrow$) | $47.87 \pm 1.95$ | $54.92 \pm 2.93$ | $52.82 \pm 3.51$ | $55.64 \pm 4.42$ |
| | F($\downarrow$) | $36.96 \pm 4.74$ | $17.24 \pm 8.46$ | $30.42 \pm 4.05$ | $9.59 \pm 6.93$ |
| RC | A($\uparrow$) | $50.42 \pm 3.92$ | $\mathbf{57.50} \pm 3.01$ | $55.93 \pm 5.85$ | $58.81 \pm 3.34$ |
| | F($\downarrow$) | $32.95 \pm 4.04$ | $\mathbf{12.91} \pm 5.00$ | $26.57 \pm 6.35$ | $8.71 \pm 5.86$ |
| EN | A($\uparrow$) | $49.18 \pm 2.36$ | $55.22 \pm 4.05$ | $53.80 \pm 4.56$ | $58.48 \pm 3.06$ |
| | F($\downarrow$) | $29.55 \pm 6.39$ | $14.95 \pm 8.85$ | $29.48 \pm 12.65$ | $8.16 \pm 3.79$ |
| BI | A($\uparrow$) | $49.18 \pm 2.61$ | $57.05 \pm 3.65$ | $50.16 \pm 2.79$ | $\mathbf{62.33} \pm 1.93$ |
| | F($\downarrow$) | $36.01 \pm 7.81$ | $13.63 \pm 8.04$ | $37.28 \pm 9.50$ | $\mathbf{7.99} \pm 4.08$ |

| KC-Cell | | | | | |
|---|---|---|---|---|---|
| Score | | M=120 | | M=160 | |
| ER | | $19.59 \pm 1.32$ | | $\mathbf{22.29} \pm 0.71$ | |
| | | $65.00 \pm 3.68$ | | $60.33 \pm 6.0$ | |
| CBR | | $18.69 \pm 1.56$ | | $20.38 \pm 1.54$ | |
| | | $66.25 \pm 2.36$ | | $62.90 \pm 4.43$ | |
| | | Top | Bottom | Top | Bottom |
| LC | A($\uparrow$) | $17.94 \pm 1.19$ | $19.66 \pm 0.83$ | $19.18 \pm 1.31$ | $21.32 \pm 1.66$ |
| | F($\downarrow$) | $67.25 \pm 3.88$ | $65.16 \pm 4.07$ | $62.90 \pm 2.48$ | $62.84 \pm 4.47$ |
| MS | A($\uparrow$) | $17.46 \pm 0.89$ | $20.05 \pm 1.79$ | $19.30 \pm 1.01$ | $21.39 \pm 0.96$ |
| | F($\downarrow$) | $66.90 \pm 3.52$ | $62.22 \pm 4.61$ | $65.13 \pm 5.05$ | $65.63 \pm 2.85$ |
| RC | A($\uparrow$) | $18.14 \pm 1.69$ | $17.83 \pm 1.20$ | $19.59 \pm 0.88$ | $21.21 \pm 1.23$ |
| | F($\downarrow$) | $66.12 \pm 3.44$ | $68.04 \pm 3.76$ | $63.50 \pm 3.76$ | $63.27 \pm 9.27$ |
| EN | A($\uparrow$) | $17.64 \pm 1.63$ | $19.46 \pm 1.54$ | $18.94 \pm 1.23$ | $21.25 \pm 1.17$ |
| | F($\downarrow$) | $68.34 \pm 4.37$ | $64.66 \pm 4.48$ | $65.57 \pm 4.24$ | $64.63 \pm 5.01$ |
| BI | A($\uparrow$) | $16.63 \pm 0.73$ | $\mathbf{20.91} \pm 1.32$ | $18.03 \pm 0.61$ | $21.61 \pm 1.23$ |
| | F($\downarrow$) | $72.63 \pm 1.45$ | $\mathbf{61.03} \pm 4.42$ | $67.90 \pm 3.89$ | $\mathbf{59.05} \pm 4.42$ |

Table 4: Comparison of average last accuracy (A) and last forgetting (F) on text classification tasks.

| Dataset | | | ER | CBR | LC | MS | RC | EN | BI |
|---|---|---|---|---|---|---|---|---|---|
| 20NewsGroup | M=60 | A($\uparrow$) | $42.33 \pm 1.66$ | $\mathbf{45.21} \pm 1.86$ | $42.90 \pm 1.23$ | $44.14 \pm 1.63$ | $44.15 \pm 1.32$ | $41.65 \pm 1.82$ | $44.92 \pm 1.63$ |
| | | F($\downarrow$) | $31.12 \pm 2.18$ | $30.39 \pm 0.80$ | $32.05 \pm 2.27$ | $31.67 \pm 2.57$ | $31.61 \pm 1.77$ | $33.77 \pm 1.43$ | $\mathbf{29.98} \pm 1.37$ |
| | M=500 | A($\uparrow$) | $45.95 \pm 1.76$ | $46.22 \pm 1.55$ | $45.72 \pm 2.28$ | $46.46 \pm 2.53$ | $46.58 \pm 2.83$ | $44.92 \pm 2.71$ | $\mathbf{46.72} \pm 2.11$ |
| | | F($\downarrow$) | $27.43 \pm 1.12$ | $28.00 \pm 0.98$ | $28.58 \pm 1.61$ | $27.97 \pm 2.36$ | $28.02 \pm 2.53$ | $29.63 \pm 2.45$ | $\mathbf{26.97} \pm 2.36$ |
| DBPedia | M=60 | A($\uparrow$) | $75.99 \pm 2.10$ | $76.24 \pm 1.21$ | $74.73 \pm 2.04$ | $75.21 \pm 1.92$ | $75.36 \pm 1.61$ | $74.58 \pm 1.88$ | $\mathbf{77.86} \pm 1.95$ |
| | | F($\downarrow$) | $24.73 \pm 2.70$ | $23.06 \pm 1.58$ | $27.06 \pm 2.30$ | $26.44 \pm 2.28$ | $26.52 \pm 1.77$ | $27.14 \pm 2.15$ | $\mathbf{23.06} \pm 2.10$ |
| | M=100 | A($\uparrow$) | $76.78 \pm 1.04$ | $\mathbf{79.78} \pm 0.83$ | $76.21 \pm 1.58$ | $76.18 \pm 1.70$ | $76.18 \pm 1.50$ | $76.33 \pm 1.45$ | $78.68 \pm 1.08$ |
| | | F($\downarrow$) | $24.14 \pm 1.18$ | $\mathbf{21.45} \pm 1.49$ | $25.15 \pm 1.60$ | $25.19 \pm 1.83$ | $25.29 \pm 1.78$ | $24.87 \pm 1.83$ | $21.79 \pm 1.41$ |
| Yahoo Answers | M=500 | A($\uparrow$) | $45.80 \pm 1.77$ | $47.45 \pm 2.86$ | $47.87 \pm 3.38$ | $47.60 \pm 3.38$ | $48.25 \pm 2.61$ | $49.32 \pm 3.61$ | $\mathbf{50.13} \pm 1.69$ |
| | | F($\downarrow$) | $40.15 \pm 2.91$ | $36.38 \pm 4.29$ | $36.02 \pm 6.81$ | $35.69 \pm 4.61$ | $34.60 \pm 3.49$ | $33.67 \pm 5.12$ | $\mathbf{32.65} \pm 3.21$ |
| | M=1000 | A($\uparrow$) | $45.48 \pm 2.11$ | $44.62 \pm 3.23$ | $46.40 \pm 3.07$ | $47.00 \pm 2.75$ | $46.33 \pm 3.68$ | $46.70 \pm 1.11$ | $\mathbf{47.03} \pm 3.36$ |
| | | F($\downarrow$) | $39.40 \pm 3.47$ | $39.17 \pm 4.16$ | $37.15 \pm 5.78$ | $37.15 \pm 4.35$ | $38.08 \pm 5.26$ | $37.85 \pm 1.78$ | $\mathbf{37.25} \pm 4.17$ |

## 5  DISCUSSION AND CONCLUSION

The novelty of the online scenario for FCL tasks raises questions on different aspects of the proposed approach and we next provide an in-depth discussion of key methodological choices and findings.

**Ablation studies - Appendix A.7.** We conduct several ablation studies to assess the impact of different hyperparameters (i.e., burn-in period, jump parameter, parameter averaging strategy, and augmentation set) on the learning performance of the proposed approach while also providing insights and explanations for usage best practices in this newly introduced context. One key aspect is the set of augmentations. From the results, the number of augmentations is important to improve the uncertainty estimation and, consequently, the performance of the approach. In terms of the choice of modality-specific augmentations, we rely on the large body of literature on TTA and show that stan-

Table 5: Comparison of average last accuracy (A) and last forgetting (F) on vision and textual tasks for standard FL, generative-based FCL, and our method.

| Method | | CIFAR10 | CIFAR100 | CRC-Tis. | KC-Cell | 20News | DBPedia | Yahoo |
|---|---|---|---|---|---|---|---|---|
| FedAvg | A($\uparrow$) | $16.90 \pm 0.55$ | $3.57 \pm 0.33$ | $22.26 \pm 2.29$ | $15.04 \pm 0.83$ | $6.34 \pm 0.58$ | $19.57 \pm 0.30$ | $14.18 \pm 1.59$ |
| | F($\downarrow$) | $78.29 \pm 2.60$ | $19.73 \pm 0.99$ | $87.39 \pm 4.06$ | $74.08 \pm 3.70$ | $58.84 \pm 1.91$ | $99.02 \pm 0.32$ | $70.90 \pm 2.88$ |
| Weighted FedAvg | A($\uparrow$) | $16.51 \pm 0.77$ | $4.22 \pm 0.20$ | $22.00 \pm 2.40$ | $14.85 \pm 0.92$ | $6.36 \pm 0.56$ | $19.23 \pm 0.44$ | $12.52 \pm 1.60$ |
| | F($\downarrow$) | $70.08 \pm 1.57$ | $20.18 \pm 0.23$ | $80.40 \pm 5.15$ | $76.21 \pm 2.72$ | $50.22 \pm 2.41$ | $97.64 \pm 1.25$ | $69.00 \pm 2.11$ |
| FedProx | A($\uparrow$) | $16.57 \pm 0.50$ | $3.73 \pm 0.15$ | $22.12 \pm 1.26$ | $14.79 \pm 1.01$ | $12.25 \pm 0.40$ | $19.58 \pm 0.26$ | $17.35 \pm 1.34$ |
| | F($\downarrow$) | $76.74 \pm 1.12$ | $20.26 \pm 0.58$ | $87.86 \pm 4.78$ | $73.38 \pm 3.66$ | $67.31 \pm 3.14$ | $99.06 \pm 0.32$ | $82.54 \pm 0.55$ |
| MFCL | A($\uparrow$) | $16.42 \pm 0.32$ | $4.60 \pm 0.09$ | $27.62 \pm 7.99$ | $16.32 \pm 1.21$ | ✗ | ✗ | ✗ |
| | F($\downarrow$) | $55.64 \pm 3.03$ | $13.69 \pm 0.33$ | $76.34 \pm 12.68$ | $70.88 \pm 1.94$ | | | |
| FedCIL | A($\uparrow$) | $16.26 \pm 1.16$ | $2.45 \pm 0.28$ | $22.44 \pm 1.23$ | $13.14 \pm 1.31$ | ✗ | ✗ | ✗ |
| | F($\downarrow$) | $49.87 \pm 0.61$ | $\mathbf{4.99} \pm 0.81$ | $62.47 \pm 1.35$ | $55.82 \pm 1.60$ | | | |
| BI (best) | A($\uparrow$) | $\mathbf{35.83} \pm 2.60$ | $\mathbf{14.31} \pm 0.76$ | $\mathbf{62.33} \pm 1.93$ | $\mathbf{21.61} \pm 1.23$ | $\mathbf{46.72} \pm 2.11$ | $\mathbf{78.68} \pm 1.08$ | $\mathbf{50.13} \pm 1.69$ |
| | F($\downarrow$) | $19.07 \pm 2.17$ | $6.96 \pm 1.40$ | $\mathbf{7.99} \pm 4.08$ | $59.05 \pm 4.42$ | $26.97 \pm 2.36$ | $\mathbf{21.79} \pm 1.41$ | $32.65 \pm 3.21$ |

dard augmentations work well both in the medical domain and for standard benchmarks of natural images. For other modalities, we propose to perform the augmentation on the latent representation level by simply adding Gaussian noise.

**Practicality and computational time - Appendices A.8 and A.9**. In terms of practicality, for standard datasets like CIFAR10 and CIFAR100, the number of communication rounds per task with a jump parameter $q = 5$ is comparable to the one reported in FedCIL (40 communication rounds per task) and considerably smaller than MFCL (100 communication rounds per task). The same consideration is also valid for the training computational time; our approach has a comparable computational time compared to FedCIL and a considerably smaller one compared to MFCL.

**Performance of generative-based approaches - Appendix A.10**. The performance gap of generative-based solutions in our experiments stems from the different assumptions of the online scenario compared to the original setting. For instance, MFCL on CIFAR100 assumes 100 communication rounds for each task. This implies that the whole task dataset is processed 100 times, allowing an effective synthetic image generation. In our online scenario, instead, we have access to one batch of 10 images at every iteration. In this case, the effectiveness of generative-based solutions is limited by their need for large amounts of data. This is in line with the online-CL literature where the most successful solutions rely on memory-based approaches (Ma et al., 2022; Soutif-Cormerais et al., 2023). To illustrate this point, we generate some images using MFCL trained in the online setting (see Figure 5 in Appendix A.10). The synthetic images provide limited information about past tasks resulting in MFCL being better than standard FL approaches without memory, but not competitive compared to our solution (see Table 5).

**Limitations.** A limitation of the proposed approach is the need to use an ensembling technique (in our case, test-time augmentation) in order to compute the BI-based estimates. This may result in a reduced efficiency compared to, e.g., ER. Furthermore, the TTA-based estimation of the Bregman Information is only an estimate of the true unknown uncertainty; the estimator used throughout the experiments (Eq. 1) is only asymptotically unbiased and underestimates the theoretical quantity (Gruber & Buettner, 2023).

**Conclusions.** In this work, we address the challenges of real-world FCL, where new data arrive in streams of small chunks that cannot be fully retained after being processed during training. While current research in FCL focuses on generative-based solutions that imply an offline setting where generators are trained at the end of each task, we advocate that in realistic conditions complete task datasets may be unavailable and local models should be updated every time new data are received. For this, we devise and formalise a new scenario for the *online* problem in FCL. To solve CF, we introduce an effective memory-based baseline that combines uncertainty-aware updates with random replay, outperforming state-of-the-art methods. Unlike generative solutions, the newly proposed BI-based estimation for epistemic uncertainty is simple to implement and applicable across different data modalities. Furthermore, compared to standard uncertainty metrics, the empirical results show its superiority in reducing CF in standard settings as well as imbalanced and probing real-world scenarios taken from the biomedical domain. This confirms the ability of the proposed uncertainty-aware strategy to sample more robust and representative samples in challenging tasks and opens new frontiers in FCL, particularly for domains where data scarcity and imbalance are prevalent.

ACKNOWLEDGMENTS

This work was supported by the The Federal Ministry for Economic Affairs and Climate Action of Germany (BMWK, Project OpenFLAAS 01MD23001E). Co-funded by the European Union (ERC, TAIPO, 101088594). Views and opinions expressed are however those of the authors only and do not necessarily reflect those of the European Union or the European Research Council. Neither the European Union nor the granting authority can be held responsible for them.

REPRODUCIBILITY STATEMENT

**Code availability.** The code used to implement our approach and to conduct the experiments is available at `https://github.com/MLO-lab/online-FCL`.

**Data accessibility.** All the datasets employed in the experiments are publicly available. We provide detailed instructions on how we use them. In case some particular preprocessing step is required, the corresponding scripts are included in our code repository.

**Hyperparameters.** All hyperparameters and training details are described in the main paper. The effect of key hyperparameter choices is investigated in the ablation study reported in Appendix A.7.

**Hardware and runtime.** As mentioned in the main paper, our experiments were conducted on a Linux machine using a single Quadro RTX 5000 and 16 GB RAM. In Tables 14 and 15, we report the expected training runtime for our approach and the mini-batch processing time in seconds respectively.

**Experiment instructions.** The code repository includes a README file with instructions to run the experiments. Furthermore, to facilitate the comprehension of our approach, Appendix A.11 describes the pseudocode of the proposed methodology.

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

## A  APPENDIX

### A.1  PREDICTIVE UNCERTAINTY SCORES

The uncertainty metrics considered in our assessment are the followings:

- *Least Confidence (LC)* (Culotta & McCallum, 2005) measures the predictive uncertainty by looking at the samples with the smallest predicted class probability. If the probability associated with the most probable class $y_{(1)}$ is low, the model is less certain about the given sample.

$$u(x)_{LC} = 1 - \frac{1}{P} \sum_{i=1}^{P} p(y_{(1)} = c|\tilde{x}_i) \tag{2}$$

- *Margin Sampling (MS)* (Campbell et al., 2000) measures the predictive uncertainty by looking at the difference between the most probable predicted class $y_{(1)}$ and the second largest one $y_{(2)}$. If the two probabilities are similar, the model is uncertain.

$$u(x)_{MS} = 1 - \frac{1}{P} \sum_{i=1}^{P} \left( p(y_{(1)} = c|\tilde{x}_i) - p(y_{(2)} = c|\tilde{x}_i) \right) \tag{3}$$

- *Ratio of Confidence (RC)* (Campbell et al., 2000) is similar to MS. In this case, instead of computing the difference, the ratio between the probabilities of the two most probable classes is considered.

$$u(x)_{RC} = \frac{1}{P} \sum_{i=1}^{P} \left( \frac{p(y_{(2)} = c|\tilde{x}_i))}{p(y_{(1)} = c|\tilde{x}_i))} \right) \tag{4}$$

- *Entropy (EN)* (Shannon, 1948) considers, differently from the previously introduced metrics, the whole probability distribution. The entropy is computed as follows:

$$u(x)_{EN} = -\frac{1}{P} \sum_{i=1}^{P} \left( \sum_{j} p(y_j = c|\tilde{x}_i) \log(p(y_j = c|\tilde{x}_i)) \right) \tag{5}$$

### A.2  SET OF AUGMENTATIONS

As described in the main paper, we employ TTA to measure the epistemic uncertainty via BI and to compute all the other uncertainty estimates. The set of augmentations used in our experiments is presented in Figure 4. Each augmentation is applied singularly on the data points of interest.

```
transform_cands = [
    CutoutAfterToTensor(args, 1, 10),
    CutoutAfterToTensor(args, 1, 20),
    v2.RandomHorizontalFlip(),
    v2.RandomVerticalFlip(),
    v2.RandomRotation(degrees=10),
    v2.RandomRotation(45),
    v2.RandomRotation(90),
    v2.ColorJitter(brightness=0.1),
    v2.RandomPerspective(),
    v2.RandomAffine(degrees=20, translate=(0.1, 0.3), scale=(0.5, 0.75)),
    v2.RandomResizedCrop(args.input_size[1:], scale=(0.8, 1.0), ratio=(0.9, 1.1), antialias=True),
    v2.RandomInvert()
    ]
```

Figure 4: Augmentation set used sequentially for the calculation of uncertainty in the experiments.

A.3 DATASET STATISTICS

In Table 6, we report the statistics of the datasets used for the main experiments. We use dataset from different domains (CIFAR10, CIFAR100, CRC-Tissue, KC-Cell) and with different modalities (20NewsGroups, Yahoo Answers, DBPedia). For CRC-Tissue, in order to create tasks with decreasing size and equal number of classes, we remove the smallest class from our evaluation. For DBPedia and Yahoo Answers Zhang et al. (2015), we randomly select a subset of 25000 and 33600 samples respectively.

Table 6: Statistics of the datasets.

| Number of | Classes | Samples | Tasks |
|---|---|---|---|
| CIFAR10 | 10 | 60000 | 5 |
| CIFAR100 | 100 | 60000 | 5 |
| CRC-Tissue | 9 | 107180 | 4 |
| KC-Cell | 8 | 236386 | 4 |
| TinyImageNet | 200 | 100000 | 10 |
| 20NewsGroups | 20 | 18828 | 5 |
| DBPedia | 14 | 33600 | 5 |
| Yahoo Answers | 10 | 25000 | 5 |

A.4 ADDITIONAL RESULTS ON TINYIMAGENET

Compared to CIFAR datasets, TinyImageNet (Le & Yang, 2015) consists of more classes (200) with larger images ($64 \times 64$ instead of $32 \times 32$), providing a slightly different scenario compared to previously explored datasets. In Table 7, we report the results for all the baselines considered in our experiments (except FedCIL which does not consider images larger than $32 \times 32$). Similar to the empirical findings reported in the main paper, BI is able to maintain competitive predictive performance while having the smallest forgetting value.

Table 7: Comparison of average last accuracy (A) and last forgetting (F) on TinyImageNet (10 tasks) for all the considered baselines.

| | A($\uparrow$) | F($\downarrow$) |
|---|---|---|
| ER | $11.26 \pm 1.31$ | $6.31 \pm 1.23$ |
| CBR | $11.06 \pm 1.65$ | $6.44 \pm 2.08$ |
| LC | $10.16 \pm 0.63$ | $6.96 \pm 1.20$ |
| MS | $10.84 \pm 1.08$ | $6.27 \pm 1.13$ |
| RC | $10.88 \pm 1.08$ | $6.76 \pm 1.17$ |
| EN | $10.18 \pm 1.02$ | $6.69 \pm 2.84$ |
| FedAvg | $1.74 \pm 0.30$ | $12.49 \pm 1.14$ |
| W. FedAvg | $1.20 \pm 0.29$ | $10.96 \pm 0.74$ |
| FedProx | $1.50 \pm 0.25$ | $10.98 \pm 0.51$ |
| FedCIL | ✗ | ✗ |
| MFCL | $3.60 \pm 0.49$ | $21.58 \pm 0.60$ |
| BI (ours) | $11.04 \pm 0.35$ | $6.07 \pm 0.74$ |

A.5 EVALUATION METRICS

In line with Yoon et al. (2021a) and Wuerkaixi et al. (2024), we employ the average *Last Accuracy* (A) and average *Last Forgetting* (F) – a federated adaptation of the metrics defined in Chaudhry et al. (2018a). *Last* refers to the measurement of the value at the end of the stream for all the clients. Suppose $a_k^{t,i}$ represents the accuracy of task $i$ after learning task $t$ on client $k$. The last accuracy $A_k$

at task $T$ on client $k$ is defined as $A_k = \frac{1}{T}\sum_i a_k^{T,i}$. Let $K$ represent the total number of clients. The average last accuracy is then defined as:

$$A = \frac{1}{K}\sum_{k=1}^{K} A_k \tag{6}$$

The forgetting measures the difference between the peak accuracy and the final accuracy of each task. Usually, the peak accuracy is reached when the considered task is trained. After that, since the model is prone to focus more on the upcoming tasks, we observe a performance degradation on the previous tasks. For this, we want to keep the forgetting as low as possible. The average last forgetting is defined as follows:

$$F = \frac{1}{K}\sum_{k=1}^{K} F_k \tag{7}$$

where $F_k$ is defined as

$$F_k = \frac{1}{T-1}\sum_{t=1}^{T-1} \max_{l\in\{1,\dots,T-1\}} a_k^{l,j} - a_k^{t,j}, \qquad \forall j < t. \tag{8}$$

## A.6   CLARIFICATION ABOUT FIGURE 3

The notable performance gap on the last task graphically describes what is the effect of catastrophic forgetting (CF) on the learning model. One can see how approaches without replay mechanisms (such as FedAvg, the dashed blue line in the figure) mainly focus on the last task with very good performance ($> 80\%$ accuracy) while completely forgetting the knowledge of the previous ones ($\sim 0\%$ accuracy). This effect can be mitigated with generative-based replay mechanisms (e.g., MFCL). In our case, thanks to the proposed approach, the accuracy across tasks is kept more uniform (between $20\%$ and $50\%$), leading to better average accuracy and reduced CF.

## A.7   ABLATION STUDY

In this section, we investigate the effect of various hyperparameters (i.e., burn-in period, jump parameter, parameter averaging strategy, set of augmentations, and number of tasks) on the learning capabilities of the proposed approach while also providing insights and explanations for best practices to use them in this newly proposed context.

**Burn-in period.** From the results reported in Table 8, the burn-in period seems to provide some benefit compared to the standard case (burn-in equal to 1) although its contribution is not crucial for improving the overall performance of the approach.

Table 8: Effect of the burn-in period on accuracy (A) and forgetting (F).

| Burn-in | A($\uparrow$) | F($\downarrow$) |
|---|---|---|
| 1 | 23.77 | 29.05 |
| 10 | 25.64 | 29.64 |
| 30 | 24.89 | 35.77 |
| 50 | 22.43 | 34.71 |
| 100 | 24.48 | 33.20 |

**Parameter averaging strategy.** The parameter averaging strategy can be changed without affecting particularly the performance of the approach as can be seen in Table 9. Overall, taking the average between the current and the previous global parameters (as proposed in Shenaj et al. (2023)) helps in reducing CF compared to the vanilla FedAvg.

Table 9: Effect of the parameter averaging strategy on accuracy (A) and forgetting (F).

| Strategy | A(↑) | F(↓) |
|---|---|---|
| FedAvg | 26.73 | 44.38 |
| Weighted FedAvg | 24.89 | 37.77 |
| FedProx | 26.67 | 38.32 |

**Jump parameter.** A value between 3 and 5 is a good trade-off for communication efficiency and good predictive performance. If the communication frequency is troublesome due to hardware constraints, one might use higher values at the price of losing pure predictive performance.

Table 10: Effect of the jump parameter $q$ on accuracy (A) and forgetting (F).

| $q$ | A(↑) | F(↓) |
|---|---|---|
| 1 | 21.15 | 16.55 |
| 3 | 37.83 | 34.84 |
| 5 | 24.89 | 35.77 |
| 10 | 22.75 | 46.46 |
| 50 | 21.99 | 37.86 |

**Set of augmentations.** To ablate the influence of the augmentations on the performance of our approach, starting from the set of augmentations listed in Appendix A.2, we randomly remove 4 and 2 augmentations from the list. The results reported in Table 11 suggest that removing augmentations decreases the performance of the proposed approach. This is probably because the higher is the number of augmentations used, the more precise is the uncertainty estimation resulting in better samples stored in the memory.

Table 11: Effect of the augmentation set on accuracy (A) and forgetting (F).

| Nr. of augmentations | A(↑) | F(↓) |
|---|---|---|
| 8 | 20.84 | 40.96 |
| 10 | 22.18 | 38.03 |
| 12 | 24.89 | 35.77 |

**Number of tasks.** For this experiment, we consider CIFAR100 as it allows us to create a larger number of tasks. From the results in Table 12, the increasing number of tasks seems to mainly influence catastrophic forgetting in all considered cases. The results look reasonable since, by increasing the number of tasks, it is more difficult to retain knowledge of the initial tasks. The same trend is also observed for the baselines included for comparison, with BI consistently outperforming ER and MFCL, with large gains especially over ER for large task numbers.

Table 12: Comparison of average last accuracy (A) and last forgetting (F) when increasing the number of tasks on CIFAR100.

| Tasks | | 5 | 10 | 20 | 25 |
|---|---|---|---|---|---|
| ER | A(↑) | 10.95 | 8.59 | 9.43 | 9.09 |
| | F(↓) | 9.12 | 20.06 | 26.76 | 41.56 |
| MFCL | A(↑) | 4.60 | 4.43 | 3.98 | 3.58 |
| | F(↓) | 13.69 | 18.59 | 31.60 | 37.43 |
| BI (ours) | A(↑) | 14.31 | 14.56 | 14.46 | 13.71 |
| | F(↓) | 8.96 | 14.36 | 19.94 | 33.65 |

## A.8 PRACTICALITY ANALYSIS - NUMBER OF COMMUNICATION ROUNDS

For standard datasets like CIFAR10 and CIFAR100, the number of communication rounds per task with a jump parameter $q = 5$ (as used in the main experiments) is comparable to the one proposed in FedCIL (40 communication rounds per task) and considerably smaller than MFCL (100 communication rounds per task). More in details, for each of the aforementioned datasets, each client possesses roughly 2000 images. With a batch size equal to 10, this results in 200 batches per task. Thus, removing the burn-in period of 30 batches, we are left with 170 batches. This means that, for a given task, the number of communication rounds with $q = 5$ is approximately 35. The frequency can be further reduced by increasing the jump parameter with a slight degradation in the predictive performance of the model (see results in the ablation study reported in Appendix A.7 Table 10.).

Table 13: Communication rounds for FCL approaches on CIFAR10 and CIFAR100.

|              | Communication rounds |
| ------------ | -------------------- |
| MFCL         | 100                  |
| FedCIL       | 40                   |
| O-FCL (ours) | 35                   |

## A.9 TIME COMPLEXITY ANALYSIS

Table 14 reports the runtime in seconds at the end of the training procedure on CIFAR10 for FedAvg (no replay mechanism), ER (random, memory-based), CBR (class-balanced, memory-based), MFCL (generative-based), FedCIL (generative-based), and BI (uncertainty-based, memory-based). From the results, the computational time of BI is in line with most of other uncertainty-based approaches and much shorter than generative approaches. The increase in runtime from CBR to BI stems from the requirement to generate TTA images. In comparison to generative approaches, the runtime of BI is similar or, compared to MFCL, considerably shorter.

Table 14: Runtime (in seconds) at the end of the learning procedure on CIFAR10.

|           | Runtime (in seconds) |
| --------- | -------------------- |
| FedAvg    | $\sim 190$           |
| ER        | $\sim 210$           |
| CBR       | $\sim 300$           |
| MFCL      | $\sim 3840$          |
| FedCIL    | $\sim 600$           |
| BI (ours) | $\sim 660$           |

In the online scenario, the total number of samples in a dataset does not affect the scalability of an algorithm. In fact, since the method does not use the whole dataset at once but rather process mini-batches sequentially, the only important factor is the time for processing a mini-batch. For this, an important factor that may affect the computational time is the image size. Table 15 reports the effect of the image size on the time for processing a mini-batch.

Table 15: Effect of the image size on the mini-batch processing time in seconds.

| Image size       | BI (ours)    | MFCL         |
| ---------------- | ------------ | ------------ |
| $32 \times 32$   | $\sim 0.20$  | $\sim 0.40$  |
| $64 \times 64$   | $\sim 0.42$  | $\sim 0.75$  |
| $128 \times 128$ | $\sim 1.32$  | ✗            |
| $224 \times 224$ | $\sim 4.19$  | ✗            |

### A.10 PERFORMANCE ANALYSIS OF GENERATIVE-BASED MODELS

The proposed online-FCL (O-FCL) scenario is very different from the standard FCL scenario presented in, e.g., MFCL (Babakniya et al., 2024) and related works based on generative-based solutions. In the standard scenario, they assume to work on task datasets that can be iterated multiple times. For instance, let us consider MFCL on CIFAR100 in which the authors assume to have 100 communication rounds for each task. This implies that the whole task dataset is processed 100 times. Thus, after many iterations over the same task dataset, the generative model can effectively generate synthetic images for the given task. In our online scenario, instead, we only have access to one batch of 10 images at every iteration. In this setting, generative-based solutions are limited in their capabilities as they require many data points and many iterations over the training data for being effective. For this reason, we believe that the performance gap compared to the standard FCL setting is given by the inefficiency of generative models in online settings where we do not have the possibility to retrain the model on the same data multiple times. This is in line with the literature in online-CL where the most successful solutions rely on memory-based solutions (Ma et al., 2022; Soutif-Cormerais et al., 2023). To prove this point, we generated some images using MFCL at the end of the training procedure in the online setting (see Figure 5). Differently from the images reported in the original paper, we can see that no informative pattern has been discovered; the synthetic images provide limited information about the past information. This is also corroborated by the results of MFCL on the datasets for vision tasks reported in the main paper. From the results, we can see that MFCL is better than standard approaches without memory, but it is not competitive compared to memory-based approaches.

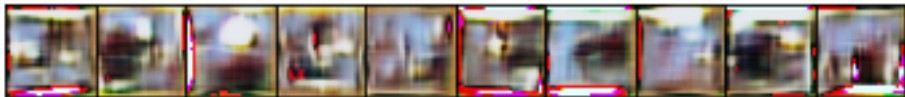

Figure 5: Synthetic images generated by MFCL (Babakniya et al., 2024) in the online setting.

A.11    PSEUDOCODE OF O-FCL

---

**Algorithm 1** O-FCL learning scheme

---

1: **Input** K: number of clients, T: number of tasks, $q$: jump parameter, *burn-in* parameter, $bs$: batch size.
2: **Notation** $k$: current client; $\mathcal{M}_k$: client memory, $[\boldsymbol{C}]$: client set, $t$: current task, $b_k^t$ current batch size for client $k$ and task $t$, $\boldsymbol{m}$: memory sample (size equal to $bs$); $bn_k^t$: number of batches for client $k$ and task $t$, $\boldsymbol{\theta}_g$: global model parameters, $\boldsymbol{\theta}_k$: local model parameters.
3: **while** training not completed for all clients **do**
4:
5:        **Client training**
6:        **for** $k \in [1, \ldots, K]$ **do**                                          ▷ For each client
7:            $b_k^t, bn_k^t \leftarrow$ GETNEXTBATCH($\boldsymbol{C}_k$)                      ▷ Get one batch from $t$
8:            **if** $t = 0$ **then**
9:                $\boldsymbol{\theta}_k \leftarrow$ TRAIN($b_k^t$)
10:           **else**
11:               $\boldsymbol{m} \leftarrow$ SAMPLEFROMMEMORY($\mathcal{M}_k$)         ▷ Random sampling from memory
12:               $\boldsymbol{\theta}_k \leftarrow$ TRAIN($b_k^t, \boldsymbol{m}$)
13:           **end if**
14:           $\mathcal{M}_k \leftarrow$ UPDATEMEMORY($\mathcal{M}_k, b_k^t$)
15:       **end for**
16:
17:       **Parameter Averaging**
18:       **if** $bn_k^t > $ *burn-in* **and** $bn_k^t \% q = 0$ **then**
19:           $\boldsymbol{\theta}_g \leftarrow$ AGGRMODEL($\boldsymbol{\theta}_k \quad \forall k \in [1, \ldots, K]$)
20:           **for** $k \in [1, \ldots, K]$ **do**
21:               $\boldsymbol{\theta}_k \leftarrow$ UPDATEPARAMETERS($\boldsymbol{\theta}_g$)
22:           **end for**
23:       **end if**
24: **end while**
25:
26: **function** UPDATEMEMORY($\mathcal{M}_k, b_k^t$)
27:       **for** $c \in b_k^t$ **do**                                        ▷ For each label $c$ in the batch size
28:           $\boldsymbol{m}_c \leftarrow \mathcal{M}_k[y = c]$                     ▷ Extract samples with label $c$ from memory
29:           $\boldsymbol{a}_c \leftarrow \boldsymbol{m}_c \cup b_k^t[y = c]$        ▷ Get candidate samples for label $c$
30:           $\boldsymbol{u}_c \leftarrow u_{BI}(\text{TTA}(\boldsymbol{a}_c))$     ▷ Compute uncertainty with Eq. (1)
31:           $\mathcal{M}_k \leftarrow$ REPLACESAMPLES($\boldsymbol{u}_c$)           ▷ Update $\mathcal{M}_k$ based on $\boldsymbol{u}_c$
32:       **end for**
33:       **return** updated $\mathcal{M}_k$
34: **end function**

---

