# OpenReview forum: "Federated Continual Learning Goes Online: Uncertainty-Aware Memory Management for Vision Tasks and Beyond"
_ICLR.cc/2025/Conference — ICLR 2025 Poster_

### Official Review · Reviewer_o8MC · 2024-11-01

**Soundness:** 3
**Presentation:** 3
**Contribution:** 3
**Rating:** 5
**Confidence:** 3

**Summary:**

The paper addresses Federated Continual Learning under the challenges of catastrophic forgetting and online learning, where new data arrives in sequential mini-batches and can only be processed once. The authors introduce an uncertainty-aware memory management strategy that leverages Bregman Information to measure predictive uncertainty. This approach selectively retains samples in memory based on their uncertainty, aiming to improve learning retention while handling various data modalities beyond vision tasks. The paper also showcases the method’s performance on vision and text data, demonstrating its capability to mitigate forgetting across diverse, real-world settings.

**Strengths:**

1. The paper presents a novel framework for Federated Continual Learning in an online setting, effectively addressing the practical limitations faced in scenarios where clients continuously receive new data without the ability to revisit previous data.

2. The innovative use of Bregman Information for uncertainty estimation allows for selective sample retention based on epistemic uncertainty.

3. The authors conduct extensive experiments with detailed analyses, demonstrating the effectiveness and depth of the proposed approach across various data types and scenarios.

**Weaknesses:**

see questions.

**Questions:**

1. The performance improvements on the CRC-Tissue and KC-Cell datasets are marginal. Given that these datasets contain more samples than others, could you provide a justification for this outcome?

2. How does the Bregman Information approach to uncertainty estimation compare with recent probabilistic methods in Federated Continual Learning, particularly regarding computational efficiency and predictive accuracy?

3. Most results are primarily compared to other uncertainty measurements, while comparisons with previous continual learning or federated learning methods appear limited.

---

> ### Author Response · Authors · 2024-11-21
> **Response to Official Review by Reviewer o8MC**
>
> Thank you Reviewer o8MC for the valuable feedback. Below are our responses to your questions.
>
> 1. **Marginal improvements on CRC-Tissue and KC-Cell**
>
> In comparison with standard class-balanced datasets of natural images, these two datasets represent a more challenging scenario. However, it is important to notice how the proposed approach has a substantial gain compared to state-of-the-art FCL methods (c.f. Table 5). As explained in response to _Reviewer 2eAg_, the lower forgetting of FedCIL on KC-Cell (and CIFAR100) is due to the very poor predictive performance in terms of accuracy. Being CF directly related to the peak accuracy for each task, the resulting value will be low if the peak accuracies are low.
>
> 2. **Comparison between Bregman Information and probabilistic methods in FCL**
>
> Could you please specify the works you are referring to? We are happy to include any baselines from this area that we may have missed.
>
> 3. **Limited comparison with previous CL and FL methods**
>
> Our contribution, investigating online FCL, is rooted in the federated continual learning (FCL) literature. With our submission, we build on the growing body of literature that investigates FCL and, itself drawing from both CL and FL literature, has established a new state-of-the-art. We therefore focus our empirical investigation - also to allow for continuity and and fair comparisons - on methods benchmarked in the FCL literature. In addition, we provide an idea of how standard CL and FL baselines would work by including ER and CBR (as a reference for CL methods) and different FL strategies (FedAvg and FedProx). This is in line with previous FCL works (e.g., FedCIL [1] and MFCL [2]). We argue that broader comparisons with other CL methods that have not shown any promise in FCL (and would typically require substantial adjustments to work in the online and federated setting) are out of the scope of this paper.
>
> [1] Qi, Daiqing, Handong Zhao, and Sheng Li. "Better generative replay for continual federated learning." ICLR 2023.
>
> [2] Babakniya, Sara, et al. "A data-free approach to mitigate catastrophic forgetting in federated class incremental learning for vision tasks." Advances in Neural Information Processing Systems 36 (2024).
>
> [3] Soutif-Cormerais, Albin, et al. "A comprehensive empirical evaluation on online continual learning." Proceedings of the IEEE/CVF International Conference on Computer Vision. 2023.

---

### Official Review · Reviewer_BinS · 2024-11-03

**Soundness:** 3
**Presentation:** 2
**Contribution:** 2
**Rating:** 6
**Confidence:** 4

**Summary:**

The paper proposes a federated learning with end points doing online continual learning (CL), called online-FCL. To address the new task, the paper proposes a memory based CL method that manage samples based on its uncertainty score using Bregman Information (BI). The proposed method is evaluated on CIFAR10/100. On vision, medical and textual datasets, the proposed method does not show much of gain but in multi-modal data, the proposed method improves the performance over the state of the arts by significant gains.

**Strengths:**

* Proposing a new federated learning with **online** continual learning.
* Proposed method improves the accuracy significantly over the state of the arts in multi-modal (vision-and-textual task) setup

**Weaknesses:**

* The proposed method is a straightforward application of Gruber & Buettner (2023), thus the technical novelty is limited.
* Empirical gain seems marginal (also considering the standard deviation) compared to prior arts (Table 1-4). But in Table 5, the empirical gain in vision and textual task seems significant. Any reasoning for this?
* Empirical validation is limited due to the size of the dataset. Although CIFAR-10/100 are popularly used datasets in CL literature, they are quite small sized. ImageNet-1K would be the minimum large scale dataset to validate the value of the proposed method.
* Some results are not clear
  * In Fig. 3, why the BI and MFCL improve the last accuracy significantly only at the last accuracy?
* In L291-292, it is claimed that "the BI-based estimation of the epistemic uncertainty is meaningful also under distribu- tion shift and able to identify robust and representative samples". But is this fact used in your method? If so, can you analyze that this is true in the Fed CL context?
* Presentation can be improved.
  * In several lines, line break has been removed -- L056, L271, L290 to name a few
  * Some abbreviation is not used without definition -- L072 CF -> catastrophic forgetting (this should be defined in L048)
  * Figure 1 could be larger and clearer. Currently it is drawn with too small fonts, making visibilty bad.
* Misc.
  * Similar previous work that uses CE loss for uncertainty of a sample is missing: Koh et al., **Online Boundary-Free Continual Learning by Scheduled Data Prior**, *ICLR 2023* (the method called CLIB)

**Questions:**

Please refer to my comments in the weakness section.

======== Justification for my final rating after the discussion with authors ================
The authors argue the technical novelty of the method of using BI for online continual learning and I appreciate/agree to it. While the technical novelty is acknowledged, the empirical validation with a small scale dataset still limits the value of the work. But with the novelty issue has been resolved, I raised my rating to borderline accept as my final rating.

---

> ### Author Response · Authors · 2024-11-21
> **Response to Official Review by Reviewer BinS (1/3)**
>
> Thank you Reviewer BinS for your thorough and helpful comments. Below are our responses to your concerns.
>
> 1. **Technical novelty is limited**
>
> We agree that the *technical* novelty is limited as we indeed build on the technical contribution from [1] who introduce a general bias-variance decomposition for strictly proper scores, and identify the Bregman Information as the variance term. However, it is widely acknowledged in the machine learning community that the notion of novelty does not originate only from such theoretical innovations. Novelty could also come from proposing new scenarios, shedding light on overlooked aspects, drawing new connections between research areas and applications, and so on. In this spirit, we argue that our realisation that (i) in the current (F)CL literature uncertainty-based memory management relies on measures of aleatoric uncertainty and (ii) that BI can be interpreted as a measure of epistemic uncertainty that can efficiently be estimated via TTA is indeed novel ([1] is a theoretical contribution with only very limited experiments). Furthermore, we would like to emphasize that the novelty of our submission does not only stem from the introduction of TTA-based BI as an uncertainty measure in online FCIL. More specifically, novel aspects of our paper include that we (i) formalise online federated CIL as a practically relevant generalisation of FCIL; (ii) demonstrate that state-of-the-art generative approaches such as MFCL and FedCIL adapted from offline federated CIL to online federated CIL do not work in the online setting and discuss the reasons for the observed performance gap; (iii) show that uncertainty-based approaches consistently outperform generative approaches for this task; (iv) point out that most of the uncertainty metrics mostly capture the irreducible aleatoric uncertainty and hypothesize that it may be more beneficial to focus on the epistemic uncertainty, and (v) introduce the use of the Bregman Information as an alternative method for predictive uncertainty estimation in the context of memory-based methods – which is largely unexplored in both federated and non-federated settings.
>
> 2. **Clarification on empirical gain between Tables 1-4 and Table 5**
>
> A key insight of our work is that state-of-the-art offline FCL baselines (MFCL and FedCIL) and standard FL (FedAvg, weighted FedAvg, and FedProx) are substantially and systematically outperformed by memory-based solutions when moving from the offline FCL setting to the online FCL setting – this is shown in Table 5. The comparison within memory-based approaches (i.e., ER, CBR) and uncertainty-based memory approaches (e.g., EN, RC) shows that BI has a competitive performance, but in comparison to the gain over SOTA FCL methods, the improvement is smaller – shown in Tables 1-4. This resulted in the following novel insights: a) generative-based solutions commonly used in offline FCL are ineffective in learning sufficiently high-quality images in the online FCL setting, where each batch can only be processed once and we don't have access to the whole task dataset multiple times; b) memory-based approaches seem the natural solution for this setting, where BI stands out for its competitive predictive performance and reduced catastrophic forgetting compared to other solutions.

---

> > ### Author Response · Authors · 2024-11-21
> > **Response to Official Review by Reviewer BinS (2/3)**
> >
> > 3. **Limited empirical validation**
> >
> > Our contribution, investigating online FCL, is rooted in the FCL literature. To facilitate a fair comparison to baselines from the FCL body of literature, we follow the experimental trend of the state-of-the-art baselines for FCL for the empirical evaluation. As can be seen in the table below, ImageNet-1K is not used. We would like to emphasize that we evaluate our approach on a larger number of datasets compared to state-of-the-art FCL baselines. The choice of those additional datasets was driven by the motivation to compile a selection of diverse datasets that reflect real-world applications and therefore include real-world medical data as well as imbalanced datasets. Our empirical evaluation demonstrates the effectiveness of our proposed approach in both standard and real-world settings.
> >
> >
> > | | MNIST | E-MNIST-L | E-MNIST-B | CIFAR10 | CIFAR100 | TINYIMAGENET | SUPERIMAGENET | CRC-Tissue | KC-Cell | 20Newsgroup | YahooQ&A | DBPedia |
> > | - | :-: | :-: | :-: | :-: | :-: | :-: |:-: | :-: | :-: | :-: | :-: | :-: |
> > | - | 60000 | 145600 | 131600 | 60000 | 60000 | 200000 | (see below) | 107180 | 236386 | 18828 | 33600 | 25000 |
> > | FedCIL            | &check; | &check; |&check; |&check; | &cross; | &cross; |&cross; |&cross; |&cross; |&cross; |&cross; |&cross; |
> > | MFCL              | &cross;| &cross;| &cross;| &cross;| &check; | &check; | &check; | &cross; | &cross; | &cross; | &cross; | &cross; |
> > | BI (ours)         | &cross;| &cross;| &cross; | &check; | &check; | &check; | &cross; | &check; | &check; | &check; | &check; | &check; |
> >
> > SuperImageNet is a dataset introduced in [2] for the offline setting by superclassing the original ImageNet. There are three versions of it; Small (S), Medium (M), and Large (L). The number of samples for each dataset is 250K, 375K, and 375K respectively. This resulted in a dataset that is a) well-balanced and b) has a large number of examples per class (spanning from 2500 in the S version to 7500 in the L version), making it particularly suitable for generative solutions in an offline setting where the model has access to all past data, but not necessarily practically relevant for online settings.
> > We believe that evaluating the proposed methodologies on more probing and realistic scenarios from other domains (such as CRC-Tissue and KC-Cell) is important to understand the validity of the approach under challenging real-world conditions (limited and different number of examples per class).
> >
> > To meet your request, we started running the experiments for TinyImagenet (which is also used in the FCL literature) to have an additional dataset in our evaluation. Below are the preliminary results. Unfortunately, since FedCIL does not use images larger than 32x32, we were not able to reproduce the results and report them on time.
> >
> > | Strategy | Acc. | For. |
> > | - | - | - |
> > | ER           | 11.26 | 6.31 |
> > | CBR          | 11.06 | 6.44 |
> > | LC           | 10.16 | 6.96 |
> > | MS           | 10.84 | 6.27 |
> > | EN           | 10.18 | 6.69 |
> > | RC           | 10.88 | 6.76 |
> > | BI (ours)    | 11.04 | 6.07 |
> > | MFCL         | 3.60 | 21.58 |
> >
> >
> > 4. **Clarification about Figure 3**
> >
> > As explained in response to *Reviewer hU4Ev*, the notable performance gap on the last task graphically describes what is the effect of Catastrophic Forgetting (CF) on the learning model. In general, the learning model tends to mainly focus on the last task. This effect is pronounced for approaches without replay mechanisms (such as FedAvg) while it can be mitigated with generative-based replay (such as MFCL) and even more with our approach.
> >
> >
> > 5. **Bregman Information under distribution shift**
> >
> > The statement (and choice of BI) was motivated by our interpretation of the online-FCL scenario as a special case of distribution shift: with the model being exposed to new classes over time, it is effectively faced with a distribution shift every time a new task is introduced. We believe that, given the robustness of BI, its use in the early stage of the training phase on a new task may result in a better sample selection compared to traditional uncertainty-based strategies - this is in line with the empirical evaluation of BI for selective classification under distribution shift in the primary publication.

---

> > > ### Author Response · Authors · 2024-11-21
> > > **Response to Official Review by Reviewer BinS (3/3)**
> > >
> > > 6. **Presentation can be improved**
> > >
> > > Thanks for pointing out these improvements. We will make sure to include them in the revised version of the paper.
> > >
> > > 7. **Comparison with CLIB [3]**
> > >
> > > Thanks for pointing out this reference. The work uses the CE loss to evaluate the importance of samples in the memory. However, this memory management strategy is tailored to the very specific scenario of online continual learning on class incremental blurry task configuration introduced in the paper. In fact, from the results reported in Table 2 of [3], CLIB has merely comparable performance with popular CL baselines in the standard online-CL scenario. For this reason, as also explained in response to other reviewers, we include ER and CBR as strong baselines for standard memory-based approaches in online CL [4].
> > >
> > > [1] Sebastian Gruber, Florian Buettner. "Uncertainty Estimates of Predictions via a General Bias-Variance Decomposition." Proceedings of The 26th International Conference on Artificial Intelligence and Statistics, PMLR 206:11331-11354, 2023.
> > >
> > > [2] Babakniya, Sara, et al. "A data-free approach to mitigate catastrophic forgetting in federated class incremental learning for vision tasks." Advances in Neural Information Processing Systems 36 (2024).
> > >
> > > [3] Koh, Hyunseo, et al. "Online continual learning on class incremental blurry task configuration with anytime inference." ICLR 2022.
> > >
> > > [4] Soutif-Cormerais, Albin, et al. "A comprehensive empirical evaluation on online continual learning." Proceedings of the IEEE/CVF International Conference on Computer Vision. 2023.

---

> ### Comment · Reviewer_BinS · 2024-11-29
> **Response to the authors' response.**
>
> Thank you for clarifying the points I raised. Here are my comments on your response.
>
> 1. I do acknowledge the value of the new scenarios and the value of using BI for measure of uncertainty. But as the authors also agree that the technical novelty is quite limited and it is certainly a point of weakness (but not a reason to reject the submission). And any comments on my argument that it is a straightforward application of Gruber & Buettner (2023)?
>
> 2. I appreciate the authors' intuitive explanation for the reason. But I was expecting the reasons backed by some empirical results or with some (further) quantitative analysis.
>
> 3. Thank you for the additional results. The reason to ask ImgNet-1K is for seeing the performance on larger sized image data with larger number of datapoints. With the TinyImageNet, I can see that the performance of methods in the larger number of datapoints.
>
> 4. I respectfully argue that 'a learning model' (I don't know what this means exactly) is 'mainly focusing' on the last accuracy only. If the learning method is sound, it should improves the accuracy throughout the inference process as a whole. It motivates to use metrics such as average accuracy, average anytime accuracy (AAA) [1] and $A_{AUC}$ [2].
>
> [1] Caccia et al., "Online Fast Adaptation and Knowledge Accumulation (OSAKA): a New Approach to Continual Learning", NeurIPS 2020
>
> [2] Koh et al., "Online Continual Learning on Class Incremental Blurry Task Configuration with Anytime Inference", ICLR 2022
>
> 5. "its use in the early stage of the training phase on a new task may result in a better sample selection compared to traditional uncertainty-based strategies" Do you have an evidence to claim this?
>
> 6. Thank you.
>
> 7. CLIB's memory management method [2] is applicable to any task configuration including standard (disjoint) memory-based online CL as the disjoint setup is a special case of blurry setup. So, you can compare your method to it.
>
> Overall, I think that the method is very promising but has some room for improvement. So, I keep my initial rating.

---

> ### Author Response · Authors · 2024-12-01
> **Response to Reviewer BinS**
>
> Thank you for the feedback. Here is the response to the remaining concerns:
>
> 1. We respectfully disagree that a lack of technical novelty is certainly a point of weakness. A piece of research does not have to be novel _and technical_ to be considered valid. We do rely on [1] in the sense that we use their decomposition for estimating predictive uncertainty. However, although the measure itself is not part of our contribution, we are the first to employ Bregman Information in this context (which is largely unexplored), interpreting it as a measure of epistemic uncertainty and thereby shedding light on its behaviour under different conditions than the original paper. Furthermore, we are the first to hypothesize that common uncertainty scores are not well-suited in this setting given their focus on aleatoric uncertainty. We use the Bregman Information to test this hypothesis.
>
> 2. The performance gap between memory-based and generative-based approaches stems from the quality of the samples used for replay. Memory-based approaches use original and informative samples. We do provide empirical evidence that samples generated via MFCL are not informative enough to help the model recall the past information. We show examples of the images generated by MFCL in Appendix A.10. Figure 5 shows empirically that synthetic images generated in the online scenario do not carry enough information about past tasks. This is because, in the online setting, generative models do not have the possibility to retrain the model on the same data multiple times.
>
> 4. We absolutely agree and indeed illustrate average accuracy in Figure 3 (the figure shows the accuracy of each task at the end of the learning process). For methods without replay mechanisms (such as the ones using only FedAvg), the model tends to focus only on the last task. In our case (BI), instead, the accuracy is kept uniform across all tasks also at the end of the training - or as you put it, the method is sound as it improves the accuracy throughout the inference process as a whole. This also results in a higher average accuracy (shown in the legend in Figure 3), which is also the metric we employ in all our experiments.
>
> 5. To demonstrate the effectiveness of our approach compared to traditional uncertainty-based strategies in the early stage of the training phase, we decided to perform an experiment where we stop updating the memory after a limited number of batches (50). In this way, the memory is populated according to samples encountered in the first part of the learning process. Then, the "frozen" memory is used for replay in the remaining part of the training procedure. Finally, we report the average accuracy and forgetting and compare them with the original values.
> Following our hypothesis, the memory should consist of different samples when using our approach compared to others and the higher quality of the samples in the memory should be reflected in better (relative) performance than other strategies. Below are the results. From the table, we can see how our solution has competitive performance also in this case, demonstrating its effectiveness already in the early stage of the training phase.
>
> | Nr. Batches | | All | 50 |
> | - | - | - | - |
> | RC                | A | 22.75 | 22.33 |
> |                   | F | 45.05 | 49.78 |
> | EN                | A | 20.14 | 19.76 |
> |                   | F | 55.58 | 58.85 |
> | BI (ours)         | A | 24.89 | 23.68 |
> |                   | F | 35.77 | 37.02 |
>
> 6. As already mentioned, CLIB shows merely comparable results with standard ER in the disjoint setup, with for example an average accuracy of CLIB for CIFAR10 of 77.96\% vs. ER 77.47\%. This is in line with recent empirical surveys [2] where ER is deemed as a very competitive baseline which often obtains better results than more sophisticated methods. We therefore include ER and CBR as competitive CL baselines to compare with. We respectfully argue that adding CLIB would not add any significant value to the paper.
>
> We would be happy to know in which way we can further improve the paper.
>
>
> [1] Sebastian Gruber, Florian Buettner. "Uncertainty Estimates of Predictions via a General Bias-Variance Decomposition." Proceedings of The 26th International Conference on Artificial Intelligence and Statistics, PMLR 206:11331-11354, 2023.
>
> [2] Soutif-Cormerais, Albin, et al. "A comprehensive empirical evaluation on online continual learning." Proceedings of the IEEE/CVF International Conference on Computer Vision. 2023.

---

### Official Review · Reviewer_2eAg · 2024-11-08

**Soundness:** 3
**Presentation:** 3
**Contribution:** 3
**Rating:** 6
**Confidence:** 4

**Summary:**

The paper proposes an uncertainty aware-memory-based approach for federated continual learning in an online setting where an estimator based on Bregman Information is employed to compute model’s variance at sample level. The proposed method includes predictive uncertainty-aware updated coupled with random replays.

**Strengths:**

1.	The problem statement is meaningful and not widely talked about.

2.	The proposed method works considering memory management in an uncertainty sampling setting, and handles class imbalance and data scarcity well.

3.	The performance looks promising and overall the paper is easy to read.

**Weaknesses:**

1.	The fundamental idea of memory management is based on predictive uncertainty which is highly model-dependent and widely used.

2.	While federated continual learning in online settings is not very common, the paper proposed uncertainty estimator and random sampling for replay sets are not novel.

3.	A scalability issue may arise for large datasets such as ImageNet.

4.	It’d be interesting to see how the proposed method would work if task numbers were increased (>20).

**Questions:**

In table 5, for BI method, why do you think the last forgetting(F) score is lower than FedCIL for CIFAR100? Is it related to the increased number of classes?

Please refer to weaknesses section.

---

> ### Author Response · Authors · 2024-11-21
> **Response to Official Review by Reviewer 2eAg (1/2)**
>
> Thank you Reviewer 2eAg for your detailed and helpful feedback. Below are our responses to your concerns.
>
> 1. **Predictive uncertainty is highly model-dependant and widely used**
>
> It is true that predictive uncertainty is widely used (see also response to _Reviewer BinS_). However, we would like to emphasize that most of the uncertainty metrics used in the literature capture mainly the irreducible aleatoric uncertainty; in our contribution, we hypothesize that it may be more beneficial to focus on the epistemic uncertainty. For this reason, we introduce the use of the Bregman Information as an alternative method for predictive uncertainty estimation in the context of memory-based methods – which in itself are largely unexplored in both federated and non-federated settings.
>
> 2. **Use of uncertainty estimator and random sampling for replay is not novel**
>
> The use of uncertainty-based memory management has indeed been investigated in CL (e.g., [1]). However, the majority of the uncertainty estimates used in this context mainly capture the aleatoric uncertainty. In contrast, we hypothesize that focusing on epistemic uncertainty could give a benefit for reducing CF while maintaining similar predictive performance. We then build on the theoretical contribution from [2] who introduce a general bias-variance decomposition for strictly proper scores, and identify the Bregman Information as the variance term.
> We leverage this insight and, interpreting BI as a measure of epistemic uncertainty that is statistically well grounded in a bias-variance decomposition, are the first to suggest to use the Bregman Information for memory management. Although aleatoric and epistemic uncertainty measures appear similar in principle, their behaviour is different in practice (as also graphically explained in Figure 2 of the paper). Finally, random sampling is indeed a standard choice in the literature that we use to sample from the memory for replay.
> We further discuss novel aspects of our submission beyond the introduction of TTA-based BI for memory management in response to _Reviewer BinS_.
>
> 3. **Scalability issue for large datasets**
>
> We would like to emphasize that, in the online scenario, the total number of samples in a dataset does not affect the scalability of an algorithm. In fact, since we do not use the whole dataset at once but rather process mini-batches sequentially, the only important factor is the time for processing a mini-batch. An idea of the computational time for the training procedure (in seconds) of different methods is given in Table 12 of the manuscript. Additionally, considering your question, we believe that an important factor that may affect the computational time is the image size. For this, we run an additional experiment to evaluate the effect of the image size on the time for processing a mini-batch. The table below reports the average time for processing a mini-batch for our methods and MFCL.
>
> | Image size | Runtime (in sec.) | Runtime (in sec.) |
> | - | - | - |
> | - | BI (ours) | MFCL |
> | 32x32    | ~ 0.20 | ~ 0.40 |
> | 64x64    | ~ 0.42 | ~ 0.75 |
> | 128x128  | ~ 1.32 |  |
> | 224x224  | ~ 4.19 |  |
>
>
> 4. **Experiments with increased number of tasks (T >= 20)**
>
> We ablate the effect of the number of tasks on the evaluation metrics on CIFAR100. From the results, the increasing number of tasks seems to mainly influence catastrophic forgetting in all considered cases. The results look reasonable since, by increasing the number of tasks, it is more difficult to retain knowledge of the initial tasks. The same trend is also observed for baselines, with BI clearly and consistently outperforming ER and MFCL, with especially large gains over ER for large task numbers.
>
> | Tasks | | 5 | 10 | 20 | 25 |
> | - | - | - | - | - | - |
> | ER                | A | 10.95 | 8.59  | 9.43 | 9.09
> |                   | F | 9.12 | 20.06 | 26.76 | 41.56
> | MFCL              | A | 4.60 | 4.43  | 3.98 | 3.58
> |                   | F | 13.69 | 18.59 | 31.60 | 37.43
> | BI (ours)         | A | 14.31 | 14.56  | 14.46 | 13.71
> |                   | F | 6.96 | 14.36 | 19.94 | 33.65

---

> > ### Author Response · Authors · 2024-11-21
> > **Response to Official Review by Reviewer 2eAg (2/2)**
> >
> > 5. **Clarification about forgetting of FedCIL in Table 5**
> >
> > The small forgetting value of FedCil on CIFAR100 stems from the definition of forgetting provided in Eqs. (7) and (8). By definition, the forgetting for a given task is computed by comparing the peak accuracy (which is usually reached when the considered task is trained) and the accuracy of that task at the end of the subsequent tasks. As can be seen in Table 5, FedCIL has a very poor predictive performance (reflected in a low last average accuracy value). Thus, since the peak accuracy is very low for all the tasks, the resulting forgetting is also very small.
> >
> > [1] Bang, Jihwan, et al. "Rainbow memory: Continual learning with a memory of diverse samples." Proceedings of the IEEE/CVF conference on computer vision and pattern recognition. 2021.
> >
> > [2] Sebastian Gruber, Florian Buettner. "Uncertainty Estimates of Predictions via a General Bias-Variance Decomposition." Proceedings of The 26th International Conference on Artificial Intelligence and Statistics, PMLR 206:11331-11354, 2023.

---

### Official Review · Reviewer_hU4E · 2024-11-08

**Soundness:** 3
**Presentation:** 3
**Contribution:** 3
**Rating:** 8
**Confidence:** 2

**Summary:**

This paper presents a new federated continual learning setting to deal with different modalities in the online scenario. The proposed uncertainty-aware memory-based approach uses an estimate to measure of predictive uncertainty of samples, The extensive experiments demonstrate the effectiveness of reducing the forgetting effect in realistic settings while maintaining data confidentiality and competitive communication efficiency.

**Strengths:**

- A real-world online federated continual learning setting is proposed to adapt the practical scenario where new data arrive in streams of mini-batches that can only be processed once.
- Using a bias-variance decomposition of the cross-entropy loss for classification tasks has some merits.
- Experiments are comprehensive and well-designed.

**Weaknesses:**

- Some advanced memory replay methods are lacking.
- Why do not consider some uncertainty-aware continual learning methods, such as [1] for comparison?
- Why the number of $M$ is different from different datasets?
- The improvement on the KC-Cell dataset is incremental compared to the other datasets. why?
- On the left of Fig. 3, it is better to discuss the notable performance gap between the proposed method and the others on the last task.

[1] NPCL: Neural Processes for Uncertainty-Aware Continual Learning.

**Questions:**

Please see the weakness above.

---

> ### Author Response · Authors · 2024-11-21
> **Response to Official Review by Reviewer hU4E**
>
> We thank Reviewer hU4E for the constructive and valuable comments. Below are our responses to your concerns and questions.
>
> 1. **Lack of advanced memory replay methods**
>
> Based on the results of recent empirical surveys [1], all memory-based replay methods perform very similarly to the common Experience Replay (ER). For this reason, we decided to use ER for comparison as it can be seen as the prototypical behaviour of a standard CL approach. We also included CBR to consider the data imbalance problem.
> In general, we see our contribution, investigating online FCL, as rooted in the federated continual learning (FCL) literature. With our submission, we build on the growing body of literature that investigates FCL and, itself drawing from both CL and FL literature, has established a new state-of-the-art. We therefore focus our empirical investigation - also to allow for continuity and fair comparisons - on methods benchmarked in the FCL literature. In addition, we provide an idea of how standard CL and FL baselines would work by including ER and CBR (as a reference for CL methods) and different FL strategies (FedAvg and FedProx). This is in line with previous FCL works (e.g., FedCIL [2] and MFCL [3]).
>
> 2. **Comparison with NPCL [4]**
>
> We thank the reviewer for pointing out reference [4]. However, although NPCL proposes an uncertainty-based approach for CL, the method is designed for the offline setting. Adaptation to the online setting is non-trivial, since key components of NPCL are global regularisation and task-specific regularisation, both of which require access to past data that is not available in the online setting.
> In general, we argue that broader comparisons with other CL methods that have not shown any promise in FCL (and would typically require substantial adjustments to work in the online and federated setting, like NPCL) are out of the scope of this paper.
>
> 3. **Different memory sizes for different datasets**
>
> The memory size depends on the number of classes available for a given dataset. For each dataset, we compared different memory sizes (i.e., different number of samples per class in the memory) to understand the effect of memory size on the performance.
>
> 4. **Incremental improvement of KC-Cell**
>
> In comparison with standard class-balanced datasets of natural images, these two datasets represent a more challenging scenario. However, it is important to notice how the proposed approach has a substantial gain compared to state-of-the-art FCL methods (c.f. Table 5). As explained in response to _Reviewer 2eAg_, the lower forgetting of FedCIL on KC-Cell (and CIFAR100) is due to the very poor predictive performance in terms of accuracy. Being CF directly related to the peak accuracy for each task, the resulting value will be low if the peak accuracies are low.
>
> 5. **Clarifications about Figure 3**
>
> Thanks for pointing this out. The notable performance gap on the last task graphically describes what is the effect of Catastrophic Forgetting (CF) on the learning model. We can see how approaches without replay mechanisms (such as FedAvg) mainly focus on the last task with very good performance while completely forgetting the knowledge of the previous ones. In our case, thanks to the proposed approach, the accuracy across tasks is kept more uniform, leading to better average accuracy and reduced CF.
>
> [1] Qi, Daiqing, Handong Zhao, and Sheng Li. "Better Generative Replay for Continual Federated Learning." The Eleventh International Conference on Learning Representations.
>
> [2] Soutif-Cormerais, Albin, et al. "A comprehensive empirical evaluation on online continual learning." Proceedings of the IEEE/CVF International Conference on Computer Vision. 2023.
>
> [3] Babakniya, Sara, et al. "A data-free approach to mitigate catastrophic forgetting in federated class incremental learning for vision tasks." Advances in Neural Information Processing Systems 36 (2024).
>
> [4] Jha, Saurav, et al. "NPCL: Neural processes for uncertainty-aware continual learning." Advances in Neural Information Processing Systems 36 (2024).

---

> > ### Comment · Reviewer_hU4E · 2024-11-27
> > **Final rating**
> >
> > Thanks for the authors' reply! My comments have been considered. I have also gone through the other reviewers' comments and the authors' replies. Indeed, I am not an expert in this domain. Anyway, the idea looks fine to me and I choose to keep my rating! By the way, I would raise the score if the authors can demonstrate the superiority of the proposed uncertainty-aware method compared to other uncertainty-aware methods.

---

> > > ### Author Response · Authors · 2024-11-28
> > > **Additional uncertainty-aware memory-based baseline**
> > >
> > > Thanks for the feedback and for keeping the encouraging score. We considered your request and now added an uncertainty-aware memory-based baseline.
> > >
> > > As replied to reviewer 2eAg, [1] explicitly mention employing an uncertainty-based memory management. In particular, the authors employ _classification uncertainty_ to populate the memory with diverse samples. The predictive uncertainty, defined in Eq. (4) in [1], is computed as follows:
> > >
> > > $S_c = \sum_{t=1}^T \mathbf{1}_c \text{argmax}_c p(y = \hat{c} | \tilde{x}_t)$,
> > >
> > > $u(x) = 1 - \frac{1}{T} \max_c S_c$
> > >
> > > In other words, given a list of perturbations $T$, $u(x)$ represents an agreement score with respect to the perturbations $t \in T$. If all the perturbed versions of input $x$ (i.e., $\tilde{x}_t$) are predicted with the same label, then the score will be 0. Otherwise, the higher the score the more uncertain the model is about the considered sample (given the poor agreement when predicting perturbed versions of the same input).
> > >
> > > We decided to implement such a score and compare the results of this alternative score with our approach. The results are reported below.
> > >
> > > | Dataset | | CIFAR10 | CIFAR100 | TINYIMAGENET | CRC-Tissue | KC-Cell | 20Newsgroup | DBPedia | Yahoo Answers |
> > > | - | - | - | - | - | - | - | - | - | - |
> > > | Rainbow Memory [1] | A | 21.71 | 13.68  | 11.02 | 60.57 | 21.07 | 44.03 | 76.48 | 49.25
> > > |                     | F | 44.52 | 7.46 | 6.47 | 9.60 | 61.13 | 27.24 | 23.84 | 35.94
> > > | BI (ours)           | A | 24.89 | 14.31  | 11.04 | 62.33 | 21.61 | 46.72 | 77.86 | 50.13
> > > |                     | F | 35.77 | 6.96 | 6.07 | 7.99  | 59.05 | 26.97 | 23.06 | 32.65
> > >
> > > [1] Bang, Jihwan, et al. "Rainbow memory: Continual learning with a memory of diverse samples." Proceedings of the IEEE/CVF conference on computer vision and pattern recognition. 2021.

---

> > > > ### Comment · Reviewer_hU4E · 2024-11-29
> > > >
> > > > Thanks for the authors' quick experiments. I am sorry to tell you that I chose to keep my quite high rating. There is not enough evidence to show the superiority of the proposed uncertainty-aware method though the numerical results seem better than Rainbow Memory. The fundamental analysis along with more experimental comparisons with some computational performance metrics would be helpful.

---

### Author Response · Authors · 2024-11-21
**General response to reviewers**

We thank the reviewers for their detailed and thoughtful feedback. We are pleased that the reviewers appreciate the _new_ (reviewer BinS) **online**-FCL problem which represents a practical scenario (reviewer hU4E) to address the limitations faced in scenarios where clients continuously receive new data without the ability to revisit previous data (reviewer o8MC). The problem statement is deemed as "meaningful and not widely talked about" (reviewer 2eAg) and the experiments are "comprehensive, well-designed" (reviewer hU4E), and "extensive with detailed analyses" (reviewer o8MC).

We are thankful for the encouraging and constructive comments aimed at improving the quality and soundness of our manuscript. Below we summarize the key points addressed during the rebuttal period:

- **Novelty**: We further discuss novel aspects of our submission in response to reviewers 2eAg and BinS. We agree that the *technical* novelty is limited, as we indeed build on the theoretical contribution from [1] who introduce a general bias-variance decomposition for strictly proper scores, and identify the Bregman Information as the variance term. We leverage this insight and, interpreting BI as a measure of epistemic uncertainty that is statistically well grounded in a bias-variance decomposition, are the first to suggest to use the Bregman Information (that we estimate via TTA) for memory management. We would like to emphasize that it is widely accepted that novelty in science does not only stem from technical aspects, but could also come from proposing new scenarios, shedding light on overlooked aspects, drawing new connections between research areas and applications, and so on [2]. In this spirit, we argue that the novelty of our submission does not only stem from the introduction of TTA-based BI as an uncertainty measure in online FCIL. More specifically, novel aspects of our paper include that we (i) formalise online federated CIL as a practically relevant generalisation of FCIL; (ii) demonstrate that state-of-the-art generative approaches such as MFCL and FedCIL adapted from offline federated CIL to online federated CIL do not work in the online setting and discuss the reasons for the observed performance gap; (iii) show that uncertainty-based approaches consistently outperform generative approaches for this task; (iv) point out that most of the uncertainty metrics mostly capture the irreducible aleatoric uncertainty and hypothesize that it may be more beneficial to focus on the epistemic uncertainty, and (v) introduce the use of the Bregman Information as an alternative method for predictive uncertainty estimation in the context of memory-based methods – which is largely unexplored in both federated and non-federated settings.

- **Scalability**: Given the online nature of the proposed framework, the total number of samples in a dataset does not affect the scalability of the algorithm. Since we do not use the whole dataset at once but rather process mini-batches sequentially, the only important factor is the time for processing a mini-batch. To address the scalability concerns of reviewers 2eAg and BinS, we now include an additional experiment to evaluate the time for processing a mini-batch for different image sizes for our method and MFCL.

- **Clarification about Figure 3**: In response to Reviewer hU4E and Reviewer BinS, we clarify that the notable performance gap on the last task graphically describes what is the effect of Catastrophic Forgetting (CF) on the learning model. In general, the learning model tends to mainly focus on the last task. This effect is pronounced for approaches without replay mechanisms (such as FedAvg) while it can be mitigated with generative-based replay (such as MFCL) and even more with our approach.

- **Additional experiments and ablation studies**: We follow the suggestions of reviewers 2eAg and BinS and now include an analysis of the effect of the number of tasks (5, 10, 20, 25) on the performance, and an additional evaluation on the TinyImageNet dataset to further expand the number of datasets considered in our experiments.

[1] Sebastian Gruber, Florian Buettner. "Uncertainty Estimates of Predictions via a General Bias-Variance Decomposition." Proceedings of The 26th International Conference on Artificial Intelligence and Statistics, PMLR 206:11331-11354, 2023.

[2] Michael J. Black, “Novelty in Science”, https://perceiving-systems.blog/en/news/novelty-in-science, 2022

---

### Author Response · Authors · 2024-11-26
**Revised version of the paper**

Dear all,

Following the comments and suggestions in the reviews, we uploaded a new version of the paper. The revision includes the following major changes:

1. **Additional experiments and ablation studies**: following the suggestions of reviewers 2eAg and BinS, we now include an analysis of the effect of the number of tasks (5, 10, 20, 25) on the performance (in Appendix A.7), and an additional evaluation on the TinyImageNet dataset to expand further the number of datasets considered in our experiments (Appendix A.4).

2. **Clarification about Figure 3**: in response to reviewers hU4E and BinS, we add a clarification about the notable performance gap on the last task graphically described in Figure 3. Due to space constraints, the clarification is in Appendix A.6.

3. **Scalability**: to address the scalability concerns of reviewers 2eAg and BinS, we now include an additional experiment in the ablation study (Appendix A.7) to evaluate the time for processing a mini-batch for different image sizes.

4. **Presentation and minor changes**: following suggestions from reviewer BinS, we improve the presentation of the paper by fixing the abbreviation definition of catastrophic forgetting, removing the mentioned line breaks, and increasing the image size and font size of Figure 1 to increase its visibility.

---

### Meta-Review · Area_Chair_Hk43 · 2024-12-19

**Metareview:**

This paper proposes a new framework for Federated Continual Learning in an online setting. The authors introduce a mechanism to selectively retain representative samples in memory, emphasizing epistemic uncertainty. The method demonstrates competitive performance across multiple modalities, including vision and text, and effectively balances learning retention and communication efficiency.

After the author-reviewer discussion period, this paper received mixed final ratings. Two reviewers are positive towards this submission, while the other two reviewers still remain negative. Specifically, Reviewer o8MC didn't actively participate in the discussions, while Reviewer BinS confirmed they still have remaining concerns.

After checking the paper, reviews, and authors' responses, I think the overall quality of this paper is impressive. Extensive results and detailed responses are provided. Therefore, I recommend acceptance of the paper. The authors should include the additional discussions and results in the final revision. They should also revise their paper to address Reviewer BinS's remaining concerns.

**Additional Comments On Reviewer Discussion:**

After the author-reviewer discussion period, this paper received mixed final ratings. Two reviewers are positive towards this submission, while the other two reviewers still remain negative. Specifically, Reviewer o8MC didn't actively participate in the discussions, while Reviewer BinS confirmed they still have remaining concerns.

After checking the paper, reviews, and authors' responses, I think the overall quality of this paper is impressive. Extensive results and detailed responses are provided. Therefore, I recommend acceptance of the paper. The authors should include the additional discussions and results in the final revision. They should also revise their paper to address Reviewer BinS's remaining concerns.

---

### Decision · Program_Chairs · 2025-01-22

Accept (Poster)